# Emergent spatial goals in an integrative model of the insect central complex

**Roman Goulard**[1]*, **Stanley Heinze**[1], **Barbara Webb**[2]

**1** Lund Vision Group, Department of Biology, Lund University, Lund, Sweden, **2** Institute for Perception, Action, and Behaviour, School of Informatics, University of Edinburgh, Edinburgh, Scotland, United Kingdom

* romangoulard@gmail.com

## Abstract

The insect central complex appears to encode and process spatial information through vector manipulation. Here, we draw on recent insights into circuit structure to fuse previous models of sensory-guided navigation, path integration and vector memory. Specifically, we propose that the allocentric encoding of location provided by path integration creates a spatially stable anchor for converging sensory signals that is relevant in multiple behavioural contexts. The allocentric reference frame given by path integration transforms a goal direction into a goal location and we demonstrate through modelling that it can enhance approach of a sensory target in noisy, cluttered environments or with temporally sparse stimuli. We further show the same circuit can improve performance in the more complex navigational task of route following. The model suggests specific functional roles for circuit elements of the central complex that helps explain their high preservation across insect species.

**Data Availability Statement:** The figures dataset and the model python code are available at the following address: https://github.com/RomanGoulard/CXModel_SpaceRep_Data.

## Author summary

Even tiny animals with reduced neuronal resources need to solve 2 dimension spatial problems. In this paper, we modelled a neural network, based on the central complex connectivity, that sustains insect visual-guided navigation both to a landmark and following a previously learned route. This combined different features that have been previously highlighted in the insect brain, (1) an inner compass, allowing an allocentric representation of their orientation, (2) a positioning system, inherited from their ability to integrate their path, (3) a long-term memory of relevant locations in the environment, that allows insect to revisit feeders repeatedly for example, and (4) a sensory guidance system that provides a stable goal direction when a rewarded signal is provided. We combined these different circuits in an complementary fashion, suggesting a crucial role for path integration in all insect navigation, beyond simply ensuring homing behaviour. Specifically, using the capability of the central complex neuronal circuit to store and manipulate navigational vectors, our implementation encodes the allocentric position of the navigation goal by combining a sensory-based vector, directed toward a goal, and a homing vector, directed toward a stable origin. We show this improves significantly the navigation in two visual-guidance paradigms, reaching a distant recognised target and following a route based on a panoramic memory.

**Funding:** RG and SH are funded by the European Research Council under the grant agreement 101044220 (Recipient: SH) - EvolvingCircuits. SH is funded by the European Union, Horizon Europe, Project 101046790 – InsectNeuroNano. Views and opinions expressed are however those of the author(s) only and do not necessarily reflect those of the European Union or the European Innovation Council. Neither the European Union nor the European Innovation Council can be held responsible for them. The funders had no role in study design, data collection and analysis, decision to publish, or preparation of the manuscript.

**Competing interests:** The authors have declared that no competing interests exist.

# Introduction

Mobile animals need to navigate through space for a diverse range of tasks: foraging for food or hunting prey, searching for mates, avoiding danger, returning to a nest. The nature of the neural encodings and computations that underlie spatial navigation have long been a central question in cognitive neuroscience. In mammals, the discovery of place, head direction and grid cells [1, 2] has been taken to support the existence of an internal "cognitive map" [3, 4]. In insects, despite some species displaying remarkable navigational capabilities, the possibility that their relatively small brains could support a cognitive map remains a point of debate [5–7]. One way this debate could be resolved is by elucidating the actual neural mechanisms that support navigation.

During the last decade, the discovery of an internal compass (resembling head direction cells from mammals) in the central complex (CX) of insects [8, 9] has lead to a rapid advance in understanding of how insect brains encode and process spatial information (Fig 1). The CX has a crossroad position in the brain between sensory inputs and motor outputs [10] and its structure is highly conserved across species [10, 11]; with many circuit properties interpretable as functional components of navigation [12]. This includes what appears to be a complete circuit to support path integration [13]—monitoring displacement (direction and distance) over time to estimate current position relative to an origin (maintaining a "home vector") [14, 15], and guiding the animal back to the origin. In addition to this positioning system, the CX hosts circuits that effectively generate, transform and use navigation vectors (Fig 2): the projection geometry of intrinsic neurons appears well suited to compute vector rotation and vector addition. Consequently, vectorial operations allow the computation of some of the circuit properties, for example the virtual 180˚ shift observed in the FB subserves the transformation of the allocentric head direction into an egocentric representation of the insect's traveling direction [16, 17]. The same principle applied to wind compass input allows *Drosophila* to reverse their up-wind/down-wind behaviour based on the odour context [18].

An important insight common to these models is that the CX provides a geometric representation of space [7] and processes and combines spatial information through vector manipulation [19, 20]. To explain some sophisticated abilities in bees, such as novel short-cuts between food sources, and development of efficient trap-lines, i.e. the visit of multiple feeders in a predictable and often optimised sequence, it has been hypothesised that the CX also supports vector memories [7, 20]. That is, at rewarded locations, the CX may store the current state of the path integrator and selectively reload this memory to drive movement back to the same location.

At the same time, the conserved nature of the CX circuit suggests it plays a role in the spatial behaviour of all insects, not just central place foragers, species like wasps, bees or ants inhabiting a nest, for which efficient relocation of the nest and food sources by individuals is crucial to survival [21]. Indeed, path integration has also been observed, over smaller spatial scales, in *Drosophila* which do not possess a fixed nest or home [22, 23]. The existence of homing in "nomadic" insects allows them to revisit a rewarded location, i.e. a food source, while exploring the environment in search for new ones [24], optimizing the exploitation of the resources. This has been supported by the behavioural observation in *Drosophila* of a reset of the path integration origin whenever an optogenetic simulation of food reward was experienced [25]. An even more basic role for the CX that we have suggested in previous work is to enhance localisation of innate or learned attractive stimuli by creating persistence in a given motion direction relative to other cues [26], which allows noisy, intermittently available, or currently invisible stimuli to be tracked more efficiently. Insects shows particular ability to maintain a straight route using a convergence of sensory information, so-called menotaxis, that have been

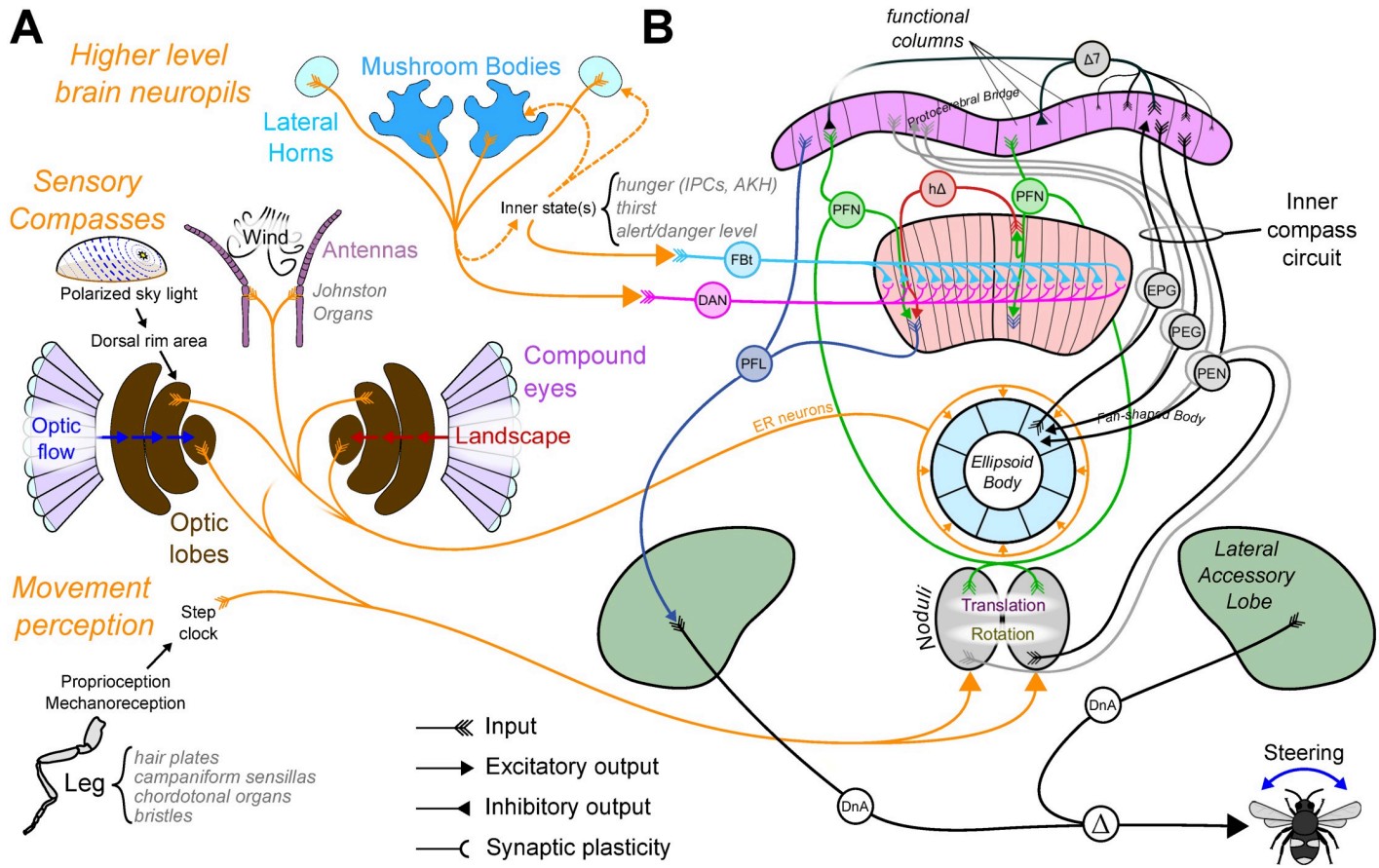

**Fig 1. Overall model anatomical diagram. (A)** Inputs to the CX from different other brain regions. We consider these inputs as three types. *Sensory Compasses* Insects are sensitive to various cues that give information about their allocentric orientation. Sensory compass pathways converge to the ellipsoid body (EB) where they connect with the *EPGs*, mostly through Ring Neurons (ER). *Movement perception* The CX receives information about self motion at the level of the Noduli (NO), mostly from visual (optic flow) and/or mechanosensory (proprioception) origin. *Higher level neuropils* The CX receives a large number of inputs from higher level regions that process multisensory information. Mushroom Bodies (MB) and Lateral Horns (LH), for example, mostly connect to the CX at the level of the fan-shaped body (FB) through FB tangential (*FBt*) neurons. **(B)** CX neuropils and the different modeled neuron type connectivity. The whole circuitry can roughly be segmented in three different functional parts, (1) the inner compass circuit (Fig D in S1 Text), represented on the top right by 4 cell types, *EPG*, *PEG*, *PEN* and *Δ7*, and projecting in-between the EB to the protocerebral bridge (PB), (2) the steering circuit (Fig 3), represented on the top left, is located between the PB, the FB and the lateral accessory lobes (LAL), including mainly 3 cell types, *PFN*, *hΔ* and *PFL*, and (3) a long-term vector memory circuit (Fig 4), represented by the parallel neural types *FBt* and *DAN* (Dopaminergic neurons) presenting a tangential projection pattern across the whole FB functional columns.

related to the CX processing [27, 28]. The ability of the CX to generate and maintain navigation vectors to sustain an oriented behaviour even in absence of new sensory information is particularly adapted to support menotactic behaviours. It is, in addition, supported by the wide range of sensory streams converging to the CX, particularly in the fan-shaped body (FB) [15, 29–31]. This CX substructure has been proposed to be the centre of the comparison between the insect's own orientation (compass) and its current goal orientation (Fig 2). The coexistence of multiple directional vectors, potentially representing competing goals, in the CX raises the question of their interactions together and with PI [32–34] to generate consistent and optimized behaviour [35].

In the current work we consider whether the anatomy of the central complex could support smooth interaction of these various forms of vectors to improve spatial navigation. More specifically, we explore whether visually guided navigation (to a visual target, or along a familiar

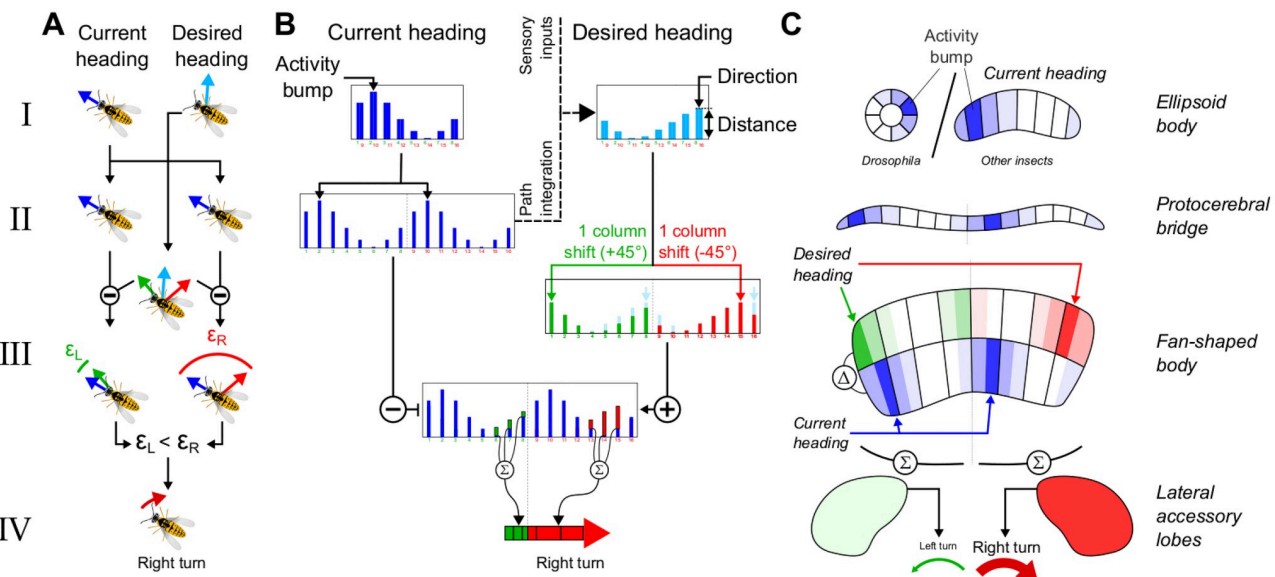

**Fig 2. Functional principle of the CX steering circuit. A-C Different model abstraction levels.** (A) Vector operations underlying the transformation of the current and desired headings into a steering output signal. (C) Neurons population activity through the different CX neuropils involved in the steering circuit. (B) Mathematical principles of the transformation of the neuron population activities into a steering output signal. **I-IV Steps of the transformation from a heading and desired heading to the steering output signal.** (I) Compass level. The neuron activity has a bump at the level of the ellipsoid body that follows the allocentric orientation of the agent. This bump, as well as the bump representing the goal orientation, take the form of a sinusoid in the model, where the peak position and amplitude can be interpreted as respectively the direction and length of a vector. (II) Compass signal copy. At the level of the protocerebral bridge the compass signal is copied into 2 corresponding hemispheric signals. This allows idiothetic rotation cues to move the compass (see Fig D in S1 Text) and, downstream, a comparison of the compass with the goal direction signals. (III) Compass and goal direction comparison. At the level of the fan-shaped body, the two copies of the compass signal are compared with rightward and leftward shifted copies of the desired heading. This allows the identification of the required direction of rotation to align with the desired orientation. (IV) Left and right hemisphere comparison. The summation of resulting population activity in each hemisphere, at the level of the lateral accessory lobes, results in a differential signal (left-right) that can be used to generate the appropriate turn toward the desired orientation. *Clipart(s) in the figure have been modified from* https://openclipart.org/.

route) can be enhanced by CX memory, using an anatomically plausible circuit model that incorporates new connectivity data. The key idea is that progress towards a goal in noisy, cluttered environments or with temporally sparse information can be scaffolded by memory of the PI location and heading direction when the insect last knew it was 'on course', which together produce an emergent internal goal. As such, this is a generic mechanism for enhancing spatial behaviour that could be common to a wide range of insects, yet easily co-opted to support the more sophisticated navigation of central place foragers.

## Model

### Overview and biological motivation

The CX model implemented for this paper builds on our previous work [26]. The model and its biological counterparts are shown in Fig 1. A core component of the model is constituted by the inner compass system in the insect brain [11]. This circuit in the CX (specifically in the EB, see *Compass circuit* section) is composed of four neuron subsets called *EPGs*, *PEGs*, *PENs* and *Δ7* (see Table 1 for name correspondence with other insects species) constituting a so-called ring attractor [10], i.e. a circuit that generates a stable representation of a circular variable, here the immediate heading direction. This head direction circuit generates an activity

**Table 1. Interspecies central complex neuropil and neuron names correspondence.** Originally from Sayre *et al.*, 2021 [36].

| Neuropils | | Columnar neurons | | FB interneurons | | Tangential neurons | |
|---|---|---|---|---|---|---|---|
| *Drosophila* | Other insects | *Drosophila* | Other insects | *Drosophila* | Other insects | *Drosophila* | Other insects |
| EB | CBL | *EPG/PEG* | $CL1_{a/b}$ | $h\Delta$ | *Pontines* | $\Delta 7$ | *TB*1 |
| PB | PB | *PEN* | CL2 | $v\Delta$ | *unknown* | *FBx* | TU |
| FB | CBL | $PFL_{1,3}$ | $CPU1_{1/2}$ | | | *ER* | *TL* |
| NO | NO | $PFL_2$ | CPU2 | | | *LNO* | *TN* |
| | | *PFN* | CPU4 | | | | |
| | | *PFR* | *unknown* | | | | |
| | | *PFG* | *unknown* | | | | |
| | | *FX* | *CU* | | | | |
| | | *unknown* | PF1 | | | | |
| | | *unknown* | $PFx_{1,2,3,4}$ | | | | |
| | | *unknown* | *PFLx* | | | | |

'bump', in the form of a sinusoidal activity pattern across the population of neurons (*EPGs*) that tracks the animal's allocentric heading [9, 37].

To generate the 'bump', we assume an accurate perception of orientation is provided by the multiplicity of sensory pathways in insects that detect directional cues [38–40] (Fig 1A). In insects these include visual cues, both from the sky (sun [41] and polarised light [42]) and the landscape [43], and also wind cues sensed by antennal displacement [44–46]. This information is carried to the EB by so-called ring neurons (ER), their topology forming plastic inhibitory synapses across the whole EB compartments [8, 43, 47]. The orientation of the different sensory compasses are dynamically imprinted, linking the egocentric orientation of a specific cue (visual, wind, . . .) with a stable inhibitory pattern, in a unified reference frame [48]. ER neurons modulate downstream *EPG* population activity to generate an integrative inner compass. In addition, the EB receives input, via the *PENs*, from sensing self-rotation through optic flow or proprioception, which can move the compass bump even in the absence of external orientation cues [49, 50].

This compass circuit then feeds into a steering circuit, which is based on a previous model of the CX as a circuit for path integration (PI) [13]. The steering circuit function is based on the phase-shift observed in the connectivity pattern of neuron groups projecting to or into the FB [16–18](Fig 1B). This particular geometry allows the comparison, at the *PFL* neurons level, of the current heading, carried by the direct inhibitory connection from $\Delta 7$ to *PFL*, with outputs from the *PFN* and $h\Delta$, both excitatory, carrying the desired heading, e.g. goal direction (Fig 2). These two pathways can be used alternatively in the model to define this goal vector based either on the head direction circuit [26] or on the PI [13]. *PFL* project onto descending neurons directly involved in the motor control of turns in *Drosophila* [51]. The hypothesised ability of the CX circuit to compare heading and goal direction have been recently supported by functional neuroimaging and modeling studies [31, 52].

To form the homing vector in the model, we assume that sensory information about self-translation (from optic flow [13, 53, 54] or proprioception [55, 56]) is projected via the noduli to *PFN*[36]. Similarly to [13], we assume accumulation of this signal across a set of directionally tuned neurons forms the home vector memory, however, in contrast to the previous model, here we locate this accumulation in the $h\Delta$ neurons. As yet, there is no direct evidence for such an accumulation property in either the *PFN* or $h\Delta$ neurons, and the previous suggestion [13] that recurrent connections within a CX column might support integration has been

challenged by cold anesthesia experiments on ants [57] and dung beetles [58], which observed a memory impairment of the PI distance only, while on bumblebees no effect was observed at smaller scale [59]. Nevertheless, the FB remains the most likely location of PI.

Finally, we implement a long-term vector memory based on a previous model of the CX as a vector-based navigation center [20]. This is consistent with the projection pattern of a set of neurons, *FBt*, that are known to receive inputs from different sensory/memory streams outside the CX (including from the mushroom bodies) and project across the whole columns of the FB [36, 60]. Moreover, we assume these neurons are modulated by context and valence, and by connecting to either *PFN* or *h$\Delta$*, the system as a whole can dynamically adjust its navigation behaviour to changing needs or motivation. Recently, a complete pathway has been unravelled from the MB to the FB and functionally shown to control upwind behaviour in the presence of a specific odor [18], supporting the existence of several context-dependent goals and justifying the position of the FB as a modulator of them.

## Implementation

The model was implemented in Python 3.6. Each neuron is represented by a simple firing rate model [61]. The neuron activation rule can be of two types, either a linear function of the input (linear simple perceptron) capped between 0 and 1 or a logic function (active or inactive based on a threshold). The input function is the sum of the activity rate of the pre-synaptic neurons. The specific connectivity between neurons is defined in a connectivity matrix (Fig C in S1 Text) and the activity level at each time step is calculated by the multiplication of the activity vector of the neurons on the previous time step by this matrix:

$$N_{act}(t) = N_{act}(t-1) \, M_{CX} \tag{1}$$

With $N_{act}$ a vector containing the rate activity of all the neurons of the CX model and $M_{CX}$ a matrix of connectivity between these neurons. The connectivity matrix is derived from known connectomic data of the main CX neuron groups (Fig 1). We particularly make use of the repeated columnar organisation of the different neuron types across the CX neuropils ($EB \leftrightarrow PB \leftrightarrow FB$). Although interspecific differences exist in the CX connections, we aimed here at unraveling general function and therefore did not target a specific species connectome but rather use representative projection patterns.

**Inputs.** *Sensory compasses.* Insects brain rely on a variety of sensory inputs to generate the activity bump in the EB [8, 9] which acts as an inner compass able to sustain navigation [62]. In this model, we simplify the contribution of the different sensory compasses by having the *EPGs* receive input from Compass Units ($I_{EB}^{Co}$, for Input from the Compasses to the EB, n = 8) with activity directly based on the agent's orientation in space (Fig D in S1 Text). Each unit has a preferred orientation, homogeneously distributed every 45° to cover the entire 360° around the agent. Their activity is calculated following a winner-take-all rule, such that only the unit with its preferred orientation closest to the agent orientation is active (1) while all the other are inactive (0). Note that, with adjustment of the gain between *EPG* and $\Delta 7$ ($k_{\Delta 7}^{EPG}$), the circuit also supports a compass input as a form of a sinusoid-shaped signal (Fig E in S1 Text).

*Self-motion perception.* Insects, and animals in general, keep track of their movements, both rotations and translations, in space for different purposes [63, 64] through sensory pathways that are fairly conserved in the insect brain [65]. In this model, we represented the perception of self-motion in the insect through two independent pathways based on the agent's translational and rotational speed respectively. In practice, the agent is moving at constant speed in its heading direction, so the translational speed pathway consists of a single neuron ($I_{NO}^{TS}$, for Inputs of the Translational Speed to the Noduli) with a constant activity, which

projects to both noduli and synapses with the *PFNs* ($I_{NO}^{TS} \rightarrow PFN_{1-16}$). The rotational speed is variable and is segmented laterally into two inputs, one for each hemisphere (Fig D in S1 Text). The activity of the neurons ($I^{TR}_{NO}$, for Inputs of the Rotational Speed to the Noduli) is binary, set to 1 (turn) in the left or right hemisphere in response to right or left turns respectively. The binary nature of these inputs might impact the ability of the compass to be maintained accurately in absence of compass inputs. As we did not simulate any disturbance in the compass inputs here, the role of the *PEN* is not crucial and have been included only to match the complete head direction circuit.

**Contextual inputs.** In addition to current sensory and self-motion inputs, the model assumes that the behavioural context provides internal states that influence the CX through tangential neuron inputs to the FB (Fig 1). We treat these inputs as two independent signals: *FBt* which controls the navigational goal (e.g. home or food); and *DANs* which signal reward in a given context and control the formation of vector memory.

**Compass circuit.** The compass circuit is designed to transform external information, acquired by diverse sensory pathways described in insects, into an inner sinusoidal function representing the immediate orientation of the agent at any time. It is similar to the one we used in our previous model work [26] and is composed of 4 different cell types, *EPG*, *PEG*, *PEN* and *Δ7* (Fig D in S1 Text). The *EPG* layer receives external inputs from the sensory compass (Fig D in S1 Text). The activity rate of the different neuron types is calculated by the following set of equations:

$$
\begin{cases}
EPG_i(t) &= k^I_{EPG} I^{Co}_{EB} i(t) + k^{PEG}_{EPG} PEG_i(t-1) + k^{PEN}_{EPG} PEN_{i\pm1}(t-1) \\[2mm]
PEG_i(t) &= k^{EPG}_{PEG} EPG_i(t-1) - k^{\Delta7}_{PEG} \Delta7_{i+4}(t-1) \\[2mm]
PEN_i(t) &= k^I_{PEN} I^{TR}_{NO}(t) + k^{EPG}_{PEN} EPG_i(t-1) - k^{\Delta7}_{PEN} \Delta7_{i+4}(t-1) \\[2mm]
\Delta7_i(t) &= k^{EPG}_{\Delta7} \sum_{i-2}^{i+2} EPG_i(t-1) - k^{\Delta7}_{\Delta7} \Delta7_{i+4}(t-1)
\end{cases}
\tag{2}
$$

Where $k^i_j$ indicates the different gain parameters regulating the strength of inputs from the cell type *i* to the cell type *j* (see Table A in S1 Text for the value of the different k parameters). All together, this circuit transforms the signal from a single activity bump, inherited from the sensory compass inputs into a temporally smoothed sinusoidal signal at the level of the Δ7 neurons (Fig D in S1 Text), that persists after the input is removed and can be moved by rotational self-motion perception. This inner allocentric compass is then transmitted further in the CX to the steering circuit, the *PFNs* and the *PFLs* layers by PB intrinsic neurons called Δ7 (Fig 1B).

**Steering circuit.** The steering circuit is primarily designed to compute the error between the compass and goal vectors [31, 52] and generates an asymmetrical signal that is decoded as the turning force for the agent (Fig 2). In addition, it holds the two navigation vectors, the heading and homing vectors, that are used to generate the behaviour through the vector memory (see next section). The circuit is largely inspired by the PI circuit previously described by Stone et al. [13] and is composed here of 3 neuron types (Fig 3A), *PFN*, *hΔ*, and *PFL*. Both *PFNs* and *PFLs* receive inhibitory inputs from the compass via Δ7. In addition, *PFNs* receive translational self-motion inputs from the noduli, consequently building a copy of the current compass orientation vector, modulated in amplitude by the translational speed. Note that in our model, the velocity of the agent being constant and without holonomic component, this scaling is simply a constant factor, equal on both brain hemispheres. We note that a more realistic model with holonomic motion and offset optic flow preferences [13], *PFN* activity could differ in each hemisphere and the circuit might need some adjustment to function correctly. *PFNs* connect downstream to *PFL* via two pathways (Fig 3B). The direct pathway implements

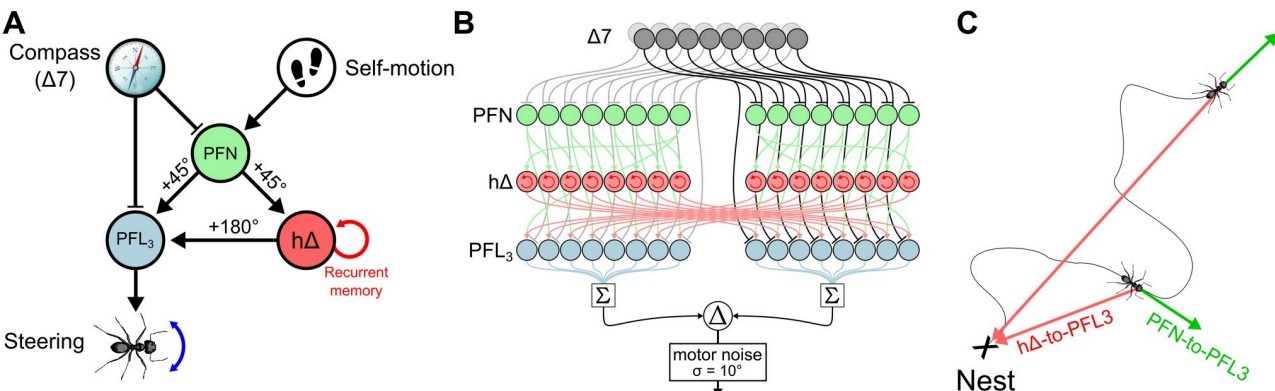

**Fig 3. Path integration circuit. (A)** Circuit diagram incorporating the direct $PFN - to - PFL$ and the indirect $PFN - to - h\Delta - to - PFL$ pathways. The activity rate of $h\Delta$ is updated continuously based on the $PFN$ inputs to retain the PI memory. **(B)** Detailed connectivity diagram of the PI circuit, from the compass circuit output ($\Delta 7$) to the steering generator (Comparison of summed outputs of the $PFL$ from both hemispheres). **(C)** Vectors encoded by the PI circuit over the path of the agent. The direct $PFN - PFL$ pathway encodes a vector of constant length and with the immediate orientation of the agent. Note that, because we did not use a purely theoretical sinusoidal signal to represent the inner compass, the $PFN$ population signal inherit some variability, in amplitude and shape, that can modify the length of this vector in a relative small magnitude. The indirect $PFN - h\Delta\text{-}PFL$ pathway encodes the integrated homing vector that points to the starting location of the path (nest/home). *Clipart(s) in the figure have been modified from* https://openclipart.org/.

a one column shift in the projection pattern. The indirect pathway, formed by $h\Delta$ neurons, receives inputs from $PFN$ and gives outputs to $PFL$ with a projection pattern producing a 180˚ phase-shift [16–18]. In addition, $h\Delta$ integrate their input, i.e. the compass orientation inherited from $PFNs$, over time, supporting a homing vector [13]. As a result $PFL$ receive inputs from the compass ($\Delta 7$) which is compared both with the current heading vector delayed by 1 simulation step ($PFN$) and the homing vector ($h\Delta$). The composition of the complete PI circuit is represented in Fig 3A. The activity rate of the different neurons of the circuit is calculated by the following set of equations:

$$\begin{cases} PFN_i(t) &= k_{PFN}^{\Delta 7}\Delta 7_{i+4}(t-1) + k_{PFN}^I I_{NO}^{TS}(t) \\[6pt] h\Delta_i(t) &= h\Delta_i(t-1) + \alpha(PFN_{i\pm 1}(t-1) - \sigma_{PFN}(t-1)) \qquad [h\Delta_i(0) = 0.5] \\[6pt] PFL_i(t) &= k_{PFL}^{\Delta 7}\Delta 7_{i+4}(t-1) - k_{PFL}^{PFN}PFN_{i\pm 1}(t-1) - k_{PFL}^{h\Delta}h\Delta_{i\pm 8}(t-1) \\[6pt] SteerCmd(t) &= k_{steer}(\Sigma PFL_{1-8} - \Sigma PFL_{9-16}) + \epsilon \end{cases} \qquad (3)$$

With $I_{NO}^{TS}$ the translational self-motion inputs from the noduli, $\alpha$ a free parameter controlling the PI memory rate, $k_{PFL}^{\Delta 7}$, $k_{PFN}$ and $k_{h\Delta}$ parameters that control the relative influence of each pathway on the steering ($k_{PFN} = 1 - k_{h\Delta} = 0.5$ by default), $k_{steer}$ a parameter that scales the steering command and $\epsilon$ a gaussian motor noise ($\sigma_\epsilon = 10$˚). $\sigma_{PFN}$ is the median activity of the whole array of $PFNs$, and is used to make the average contribution of $PFN$ activity to the PI integration zero. That is, addition to the PI sum in the heading direction is always balanced by subtraction in the opposite direction. To support this symmetric positive-negative update of the PI, the initial activity rate of the $h\Delta$ is set to 0.5 and limited to the range [0 1]. The outcome of the whole circuit is the encoding of two vectors (Fig 3B), one instantaneous orientation vector from the $PFN - PFL$ pathway, with a virtual length determined by the $PFN$ activity, itself determined by the $I_{NO}^{TS}$ (constant in our model), and another continuously updated homing vector from the $PFN - h\Delta - PFL$ pathway.

**Vector memory circuit.** The vector memory circuit is designed to store memory of either $h\Delta$, carrying the PI homing vector, or *PFN* activity rate, carrying the immediate head direction vector. A reward signal induces the copying of one or both of these vectors into memory by modulating the synaptic weight of *FBt* inputs to the corresponding neuron type ($h\Delta$ or *PFN*). The memory can then be used later, under the control of an associated contextual signal that gates *FBt* activity.

**The circuit.** We implement a similar circuit to that proposed by Le Moël et al, 2019 [20] and generalise it to support visual navigation. The model circuit is composed of two neuron types, *FBt* and $DAN_{CX}$, so-called for its hypothetical dopaminergic neurotransmitter. Both have a similar projection pattern across the functional columns of the FB and interact in similar regions of the axons of both *PFN* and $h\Delta$ cells. *FBt* cells comprise two subtypes that form inhibitory synapses with either *PFN* ($FBt^{PFN}$) or $h\Delta$ ($FBt^{h\Delta}$). In contrast, a single $DAN_{CX}$ neuron innervates both pathways simultaneously (Fig 4A) and triggers neuromodulation of the

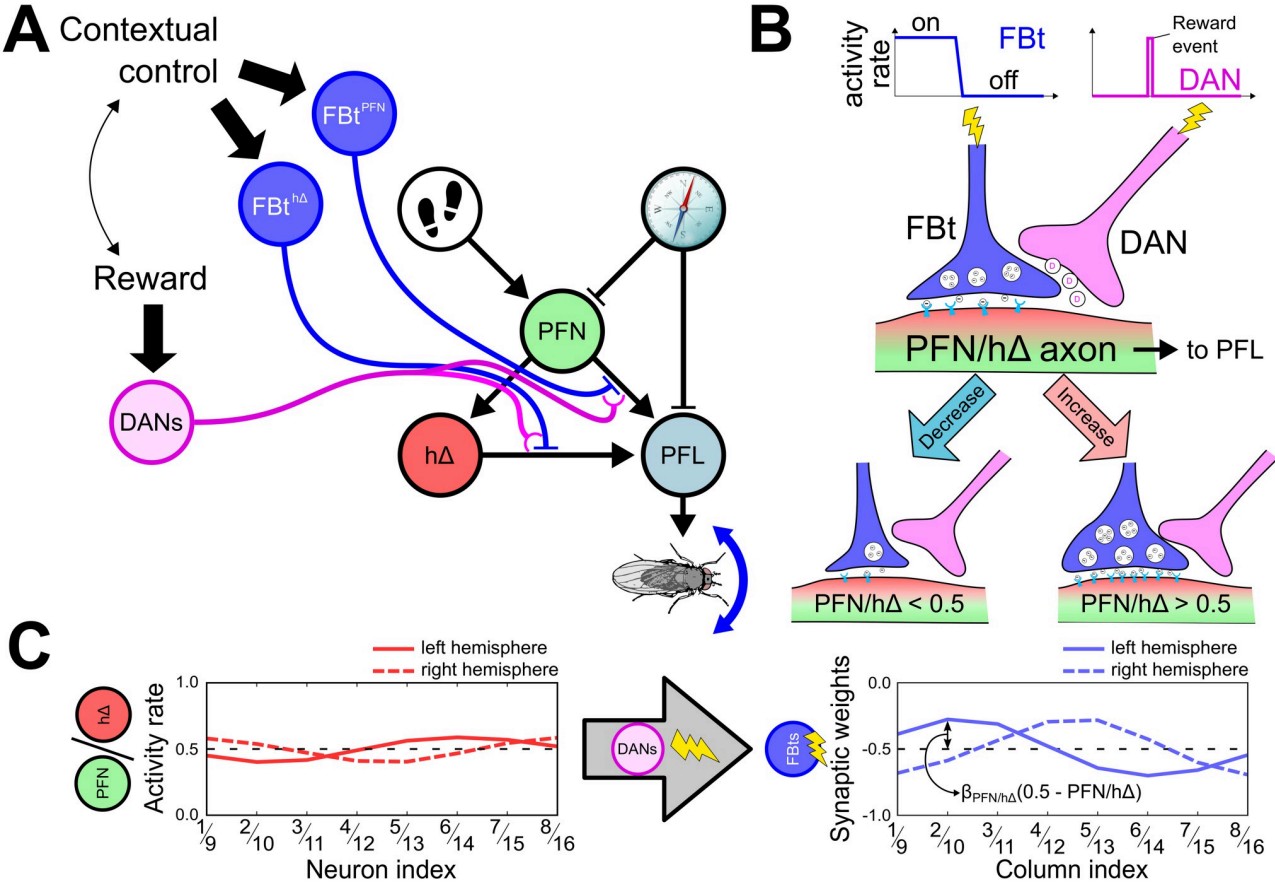

**Fig 4. Vector memory circuit. (A)** PI circuit with the addition of vector memory. The *FBt* pathways receive control from contextual/motivational signals based on the inner state (e.g. hunger). The *DANs* pathway receives inputs from reward signal(s) that define when a condition (dependent also on the context/task) is fulfilled and trigger a modulation of the $FBt - PFN$ or $FBt - h\Delta$ synaptic strength at the level of *PFN* and/or $h\Delta$ axons to *PFL*. **(B)** Proposed mechanism for vector memory by synaptic modulation within the $FBt - DAN - PFN/h\Delta$ trios. Whenever a *DAN* is activated, any active *FBt* has its synaptic strength in every column modified proportionally to the activity rate of the corresponding *PFN* or the $h\Delta$. The modulation of the synaptic weight could happen at the level of either the presynaptic partner (*FBt*), the postsynaptic level (*PFN/h\Delta*), or both. An activity rate of *PFN/h\Delta* greater than 0.5 (more active than inactive) induces an increase of the strength of the inhibitory *FBt* synapse; and an activity rate of *PFN/h\Delta* lower than 0.5 (more inactive than active) induces a decrease of the synaptic strength. **(C)** Applied to all the functional columns, this mechanism stores a copy of the *PFN* and $h\Delta$ activity rates, at the time of reward via *DANs*, in the form of altered *FBt* synaptic strengths. The amplitude of the "copy" depends on the learning rate parameters, $\beta_{PFN}$ and $\beta_{h\Delta}$. Clipart(s) in the figure have been modified from https://openclipart.org/.

$FBt - PFN$ and $FBt - h\Delta$ synapses (Fig 4A). Finally, each cell type receives different inputs, following the logic of their functionality. $FBt$ receives input from contextual or state dependent signals of the agent, whereas $DAN_{CX}$ receives input from a reward signal, also based on the agent's current state. This circuit therefore modulates the connection from both $PFN$ and $h\Delta$ to $PFL$, changing their update equations from Eq (3), to:

$$\begin{cases} I_{i\ PFL}^{PFN}(t) & = & k_{PFL}^{PFN}(PFN_{i\pm1}(t) - \sum_j [FBt_j^{PFN}\ S_j]) \\ I_{i\ PFL}^{h\Delta}(t) = k_{PFL}^{h\Delta}(h\Delta_{i\pm8}(t-1) - \sum_j [FBt_j^{h\Delta}\ S_j]) \\ PFL_i(t) = k_{PFL}^{\Delta7}\Delta7_{i+4}(t-1) + I_{i\ PFL}^{PFN}(t) + I_{i\ PFL}^{h\Delta}(t) \end{cases} \quad (4)$$

$I_{PFL}^{PFN}$ and $I_{PFL}^{h\Delta}$ represent the inputs to $PFL$ cells from, respectively, $PFN$ and $h\Delta$. Constants are inherited from Eq (3). $S_j$ represents the state j, which in the present model is limited to two different motivational states, exploration for food or returning to the nest.

*Synaptic plasticity memory.* To generate a vector memory, and more generally a goal direction, associated with a specific context or motivation, the active $DANs$ trigger the modulation of the synaptic weights between $FBt_{PFN} - PFN$ and $FBt_{h\Delta}-h\Delta$ (Fig 4B). In each column, the modulation of the synaptic strength is proportional to the activity rate of the respective $PFN$ or $h\Delta$. Because the length of the vector encoded is dependent on the population sinusoid signal (Fig 2B), we use two gain parameters, respectively $\beta_{PFN}$ and $\beta_{h\Delta}$, to modulate the amplitude of the memory encoded at the level of the $FBt$ synapses. These parameters can be set over a range of values to test their effect on the simulated behaviour. The modulation of the synaptic strength can be either positive, if the $PFN/h\Delta$ activity rate is greater than 0.5, or negative, if it is lower than 0.5. This results in a stable copy of the $PFN/h\Delta$ activity rate across the FB columns at the time of $DANs$ activation (Fig 4C) that can be used to modify the $PFN/h\Delta$ inputs to the steering neurons $PFL$ (Eq 4). To avoid any instability over time, the modulation of the memory at the synaptic level is made independent of the previous state of the synapses, i.e. the memory is flushed (the synapses are reset to their initial strength) and rewritten (the synaptic weight modulation is applied) every time the $DAN$ is activated, as follow:

$$\begin{aligned} FBt_i^{PFN}(t)\ \&\ DAN_{CX}(t) > 0 &\leftrightarrow \omega_i^{PFN}(t+1) = -0.5 - \beta^{PFN}\ (PFN_i(t) - \sigma^{PFN}(t)) \\ FBt_i^{h\Delta}(t)\ \&\ DAN_{CX}(t) > 0 &\leftrightarrow \omega_i^{h\Delta}(t+1) = -0.5 - \beta^{h\Delta}\ (h\Delta_i(t) - \sigma^{h\Delta}(t)) \end{aligned} \quad (5)$$

Where $\omega_i^{PFN/h\Delta}$ is the synaptic weight between each $FBt$ and either $PFN$ or $h\Delta$ and $\beta^{PFN/h\Delta}$ is a free parameter representing the rate of the synaptic strength modulation. $\beta^{PFN}$ and $\beta^{h\Delta}$ thus modulate the vector length kept in memory on each pathway. $\sigma^{PFN}$ and $\sigma^{h\Delta}$ are the instant mean activity level of the population of respectively $PFN$ and $h\Delta$. Thus, the term that modulates the memory is centered around 0, positive for $PFNs$ greater than the population mean activity and negative for $PFNs$ lower. Consequently, the vector memory created is centered around its initial value, represented by the constant -0.5 (inhibitory) term in the equation. This implies that the creation of a new memory wipes any previous memory, i.e. the new $FBt - PFN/h\Delta$ synaptic weights are independent from the previous ones. Note that it also keeps the location memory ($FBt - h\Delta$) consistent with the $h\Delta$ population activity (PI), centered on 0.5 in the model.

## Simulations and results

To evaluate the model in different contexts, we defined different simulation paradigms in which we observed its navigation behaviour. The simulations are done in a custom made

pyOpenGL environment (using the pyglet library). The virtual world used is composed of a blue sky, a brown ground and can either be empty or enriched by red or green objects such as vertical cylinders, cones and cubes depending on the paradigms (Fig A in S1 Text).

## Setting the model: Replication of previous implementations

**Navigation using PI and vector memory.** The first task we set consists in navigating from the nest to a known food source. It is a replication of the the work from Le Moël et al [20]. The main novelty of our implementation is the use of $h\Delta$ cells as the substrate for the PI. For this task we only focused on the vector memory of the PI and therefore omitted the direct connections from *PFN* to *PFL* in the model (Fig 5A.a). The simulations are composed of 3 parts, (1) the first outbound journey to find a food source and set the vector memory of this source, (2) the return to the nest based on the PI homing vector, and finally (3) a second outbound journey based on the interaction between the vector memory and the PI.

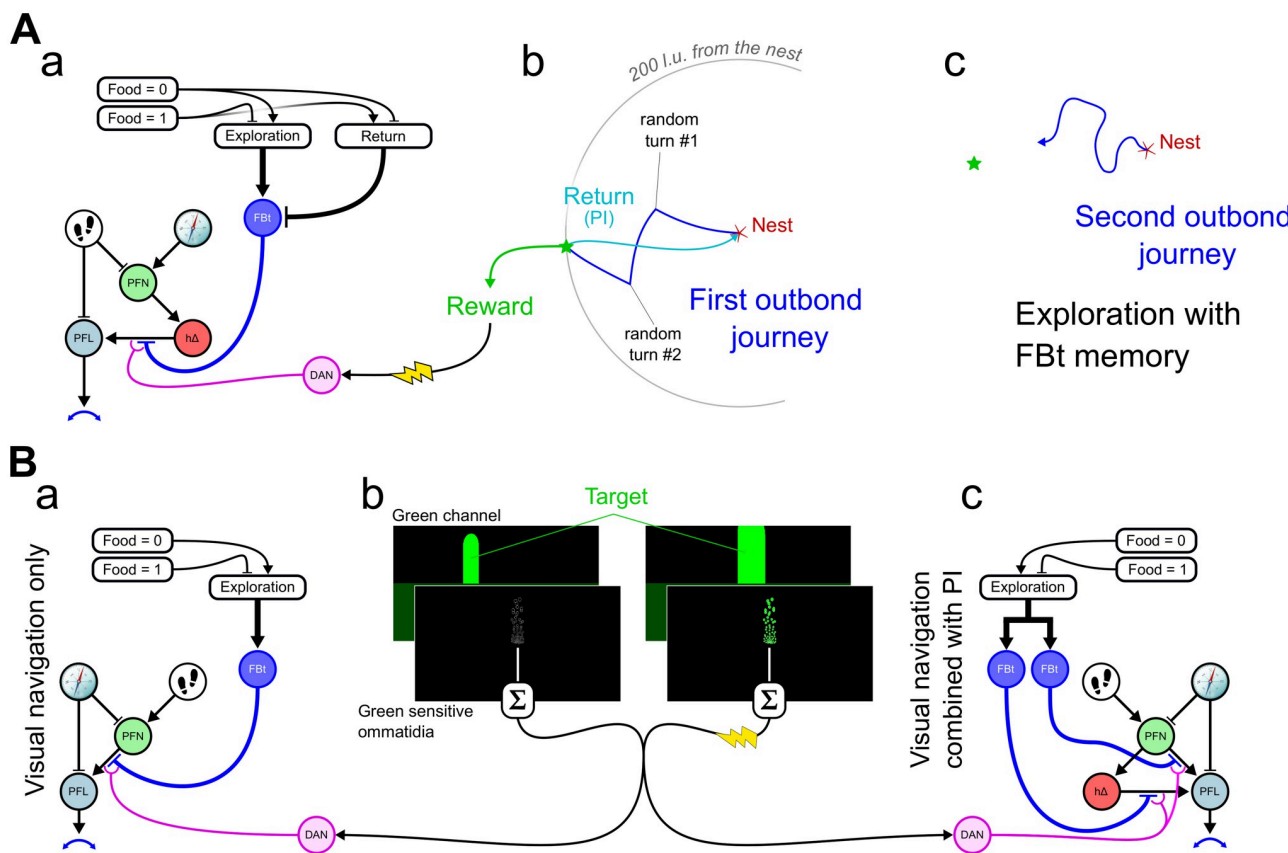

**Fig 5. Experimental paradigms. (A)** Navigation using PI and vector memory, replicating [20]. ***a.*** The steering circuit in this simulation is only composed of the $h\Delta$ pathway, supporting path integration. When the agent is lacking food (food = 0) it promotes the exploration behaviour through the excitation of a specific *FBt*. When food is found, the *DAN* circuit triggers memory formation. The motivation then switches to the return state (food = 1) where the *FBt* is inhibited, leading to homing behaviour. ***b.*** Sequence of behaviour implemented: (1) the agent leaves the nest to explore (pre-determined zig-zag pattern). (2) When the agent reach 200l.u. from the nest, it is provided with a food reward, triggering the formation of the memory while simultaneously switching the motivation to the return mode. (3) Homing behaviour to the nest (catchment area of 20l.u. diameter) ***c.*** Reset of the motivation to the exploration mode (food=0). This time, the behaviour of the agent is left under the control of the steering circuit which should bring it back to the memorised location. See results in Fig 6. **(B)** Navigation using visual guidance and vector memory, replicating and extending [26]. ***a.*** For visual navigation without PI the steering circuit is only composed of the *PFN* to *PFL* pathway. The *DAN* reward circuit receives inputs from the sum of the target-detection ommatidia activity (see B.b), so the circuit forms a vector memory of its direction when facing the target. See results in Fig 7. ***b.*** Visual circuit used to control the recognition of a green object in the central visual field (Fig B in S1 Text). ***c.*** For visual navigation with PI, the circuit includes both *PFN* and $h\Delta$ pathways to the *PFL*. The agent thus forms a memory of both the direction of the visual target and the location from which it was seen, further improving its ability to locate the target. See results in Fig 8. *Clipart(s) in the figure have been modified from* https://openclipart.org/.

To simulate the first outbound journey (Fig 5A.b), we define a Z trajectory from the nest to the food source. The agent starts with a random direction and executes two random turns in opposite directions (of respectively $100 - 120°$ and $30 - 60°$) at defined distances from the starting point ($d = 80l.u.$ & $d = 160l.u.$), generating the zig-zag pattern. When the agent reaches $200l.u.$ from the nest, we then simulate an encounter with food. The first consequence is the activation of the *DAN* pathway. This induces the synaptic change at the $FBt - h\Delta$ level and forms a vector memory at the food location. The second consequence is the activation of the return pathway, leading to a complete inhibition of the *FBt* cells. This inhibition releases the $h\Delta$ pathway, which carries the PI, and thus generates the homing behaviour. Note that we could have used the *FBt* directly to generate an outbound journey by presetting a memory to a far away location, we decided to keep the zigzag pattern instead as this gave us the opportunity as well to check on the stability of the PI memory accumulation with non-linear trajectories.

The agent then returns to the nest and it is caught when it approaches within $20l.u.$ of the nest. Once the agent has reached the nest, the $h\Delta$s activity rate is homogeneously reset to its initial value (0.5), resetting the PI. The feeding state is also reset to 0, stimulating again the exploration *FBt*.

Finally, the second outbound journey is initiated. This time the inhibition by *FBt* of the $h\Delta$ inputs to *PFL* corresponds to the vector memory formed at the food location. The agent should thus be steered to the position where its PI ($h\Delta$) activity and memory (*FBt* activity) cancel out, i.e. the food location. In order to evaluate whether a vector memory was created that included both the direction and the distance from the nest, the agent is not stopped when it reaches the $200l.u.$ limit, but rather left in exploration mode until the simulation reaches a fixed number of steps.

Fig 6A shows an example of the 2 first phases of the simulation (first outbound journey and return to the nest). The $h\Delta$ activity rate encodes correctly the home vector of the agent through

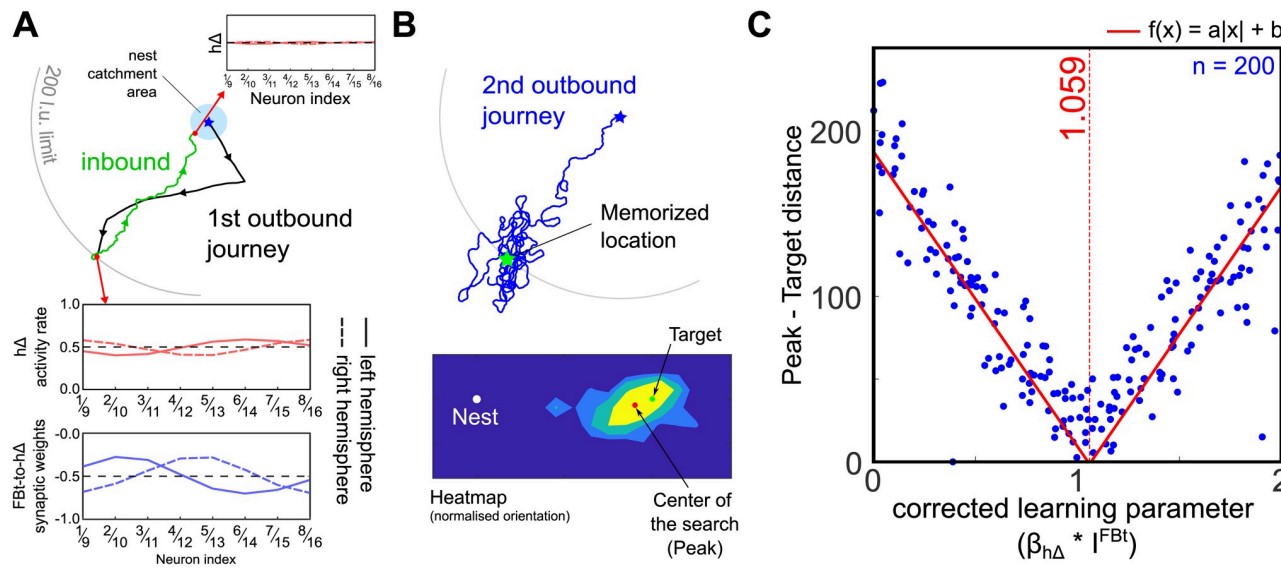

**Fig 6. PI and vector memory of a food location.** (A) PI during the outbound journey (black path) and inbound journey (green path). The activity rate of the $h\Delta$ neurons is indicated for different location of both journeys; it encodes the home vector as a sine wave with amplitude corresponding to length and phase corresponding to angle. The agent is rewarded (finds food) when it reaches a set distance from the nest (200 l.u.) and triggers the creation of the synaptic long-term memory at the level of the $FBt - to - h\Delta$ axonal connections. (B) Retrieval outbound journey. The long-term *FBt* memory induces the steering circuit to drive the agent towards the location where the memory and PI cancel, i.e., the rewarded location. The distance between the search peak and target is used as a measure of precision in C. (C) Effect of varying the learning parameter $\beta_{h\Delta}$ on the precision of retrieval journey: this modulates the vector length stored in memory. Each point represents one simulation with a fixed $\beta_{h\Delta}$ randomly chosen in the [0 4] range. Note that because we modulated the motivational input to the *FBt* with a factor $I^{FBt} = 0.5$, we corrected the value of $\beta_{h\Delta}$ to verify that the best retrieval was achieve with a perfect memory ($\beta_{h\Delta} x I^{FBt} \approx 1$).

the PI. At the location of the reward, the $h\Delta$ activity rate is efficiently copied in the synaptic weight from $FBt$ to $h\Delta$. Symmetric update of $h\Delta$ activity allows the activity to return near to its original level when it reaches the catchment area of the nest ($20 l.u.$). Subsequently, the activated $FBt$ leads the agent correctly toward the previously visited and rewarded location (Fig 6B). In addition, once it reaches the vicinity of the location, it clearly initiates a search behaviour from which we can calculate a search centre. To optimise the encoding of the vector memory, we varied the learning parameter $\beta_{h\Delta}$ in different simulations (n = 200) and evaluate the distance between the search centre and the rewarded location to estimate the influence of $\beta_{h\Delta}$. The distance between the memorised location and the search pattern centre increases linearly when $\beta_{h\Delta}$ is further from its perfect memory value ($\beta_{h\Delta} x I^{FBt} \approx 1$).

**Navigation using visual guidance.** The second task we investigated was whether the model could enhance guidance to a visual target. This replicates the task in [26]. The agent was placed in a 3D virtual world containing one target: a green vertical cylinder (Fig A in S1 Text). The key constraint on this task is that the visual system cannot estimate the target's egocentric orientation directly to drive steering, but instead can only recognise when the target is in the frontal visual field. We use an insect-inspired eye model (Fig B in S1 Text) in which a subset of ommatidia oriented toward the front (azimuth ±10˚) and above the horizon (elevation >0˚), are set to be green-sensitive: any summed activity level above 0 is considered as an alignment with the target. Target detection is used as the input to the $DAN$ reward pathway (Fig 5B). In this task, the desired vector memory is the current orientation of the agent, i.e., the direction it is facing when the target is seen. Hence, we only consider the $PFN - PFL$ pathway and ignore the $h\Delta - PFL$ pathway (Fig 5B.a). Detection of the target triggers the creation of a vector memory at the level of the $FBt$ to $PFN$ axon synapses. The virtual length of this vector is determined arbitrarily by a learning rate parameter, $\beta_{PFN}$.

Fig 7A shows 4 examples of paths using the visual guidance circuit. The model clearly succeeds in navigating in the direction of the target. However, in some cases when the target is missed the agent keeps heading in a constant direction, beyond the target (Fig 7A-*top-right panel*). The reward pathway, associated with the detection of the target in the frontal visual

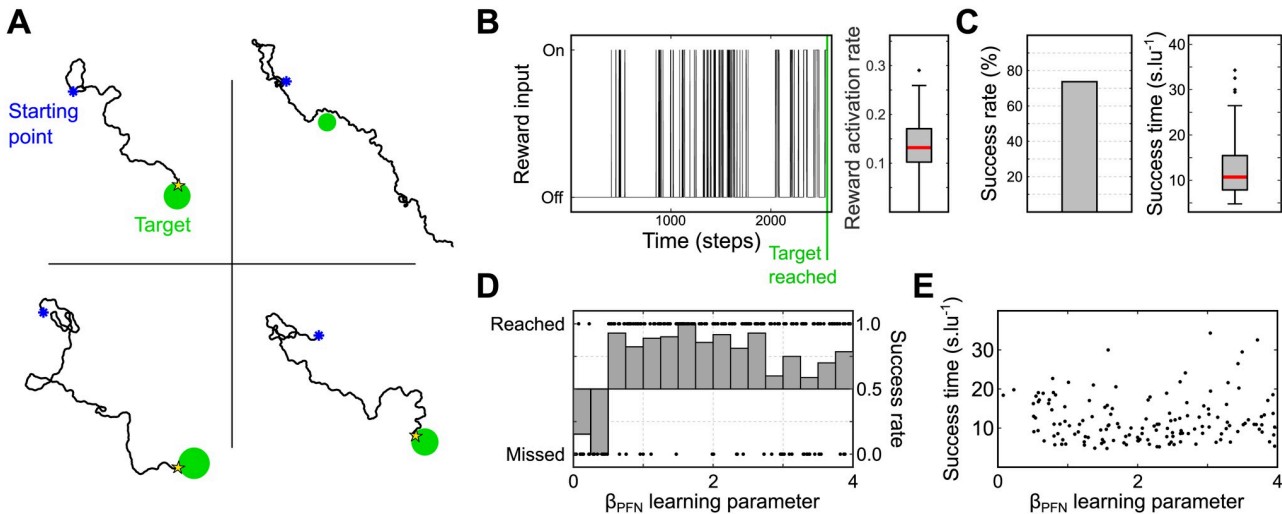

**Fig 7. Visually guided navigation.** (A) Examples of trajectories from the nest to the target. (B) *Left panel* Example of reward input activity over time steps. *Right panel* Boxplot of the reward activity rate across all simulations. (C) Success rate and time to reach the target. Time is between first sighting the target and reaching it, normalised by the distance to the target at the first sighting. (D) Success to reach the target as a function of the learning parameter $\beta_{PFN}$. (E) Time to reach the target as a function of the learning parameter $\beta_{PFN}$. *Clipart(s) in the figure have been modified from* https://openclipart.org/.

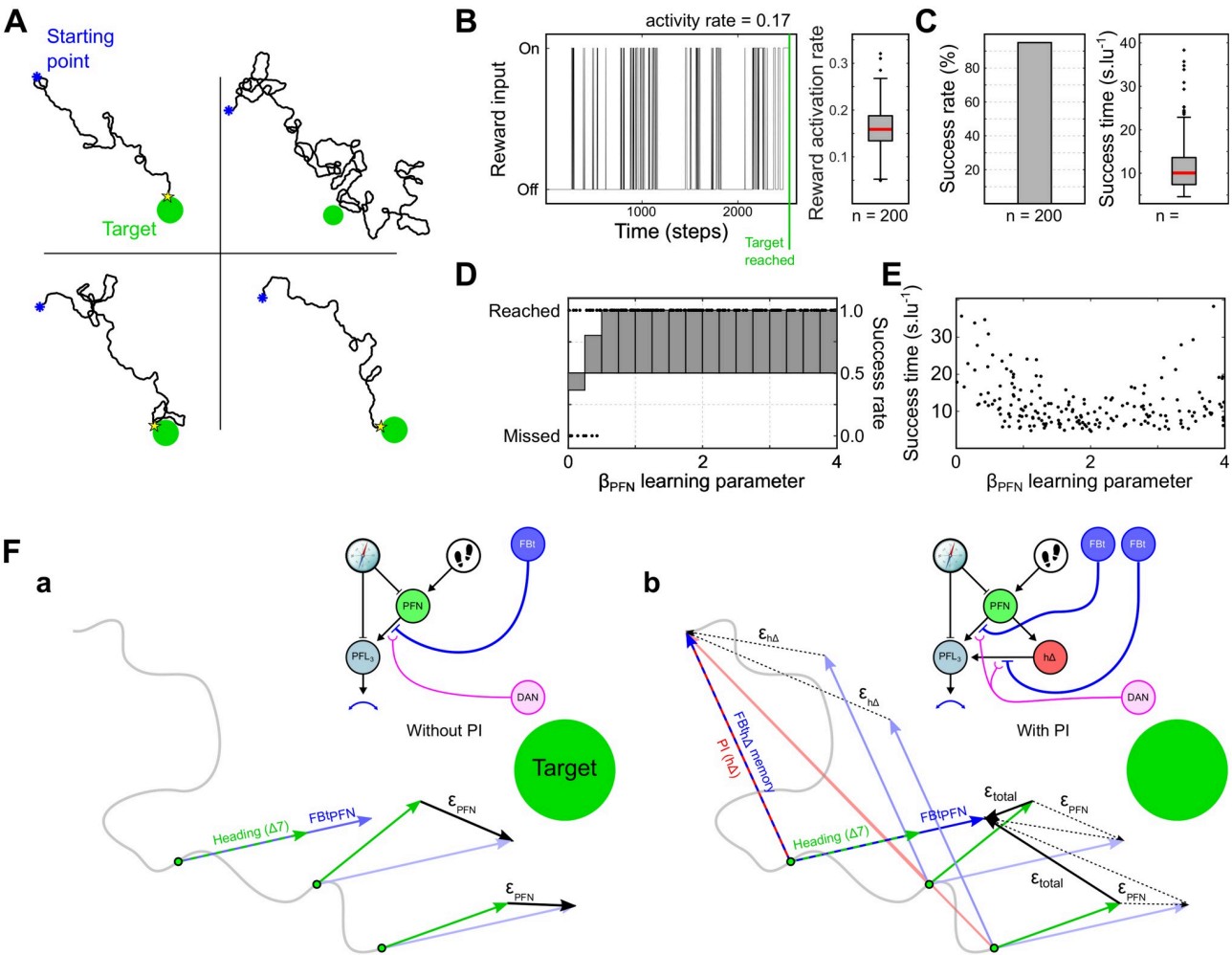

**Fig 8. Visually guided navigation including PI memory.** (**A**) Examples of trajectories from the nest to the target. (**B**) *Left panel* Example of a simulation reward input activity over time steps. *Right panel* Boxplot of the reward activity rate during the simulations (n = 200). (**C**) Success rate and time to reach the target. (**D**) Success to reach the target as a function of the learning parameter $\beta_{PFN}$. Dots indicate individual simulation outcome. (**E**) Time to reach the target as a function of the learning parameter $\beta_{PFN}$. (**F**) Conceptual comparison of the models with and without the $h\Delta$/PI pathway for visual guidance. *a.* Without $h\Delta$/PI pathway, the agent tries to correct its heading to be the same as the heading when the target was last sighted. *b.* With $h\Delta$/PI pathway, the agent is steered towards the vector sum of the location where the target was last sighted, and the heading in which it was sighted, preventing overshoot. *Clipart(s) in the figure have been modified from* https://openclipart.org/.

field, follows an intermittent pattern of activation, with a rate of activation between 0.1 to 0.2 (Fig 8B). The overall success rate to reach the target is relatively high (around 75%, Fig 7C) considering the simple target detection system. In addition, most of the unsuccessful simulations are observed with a low learning parameter $\beta_{PFN}$ (Fig 7D). It seems for any value of $\beta_{PFN}$ between 0.5 and 3 the success rate and time is similar (Fig 7D).

## Adding PI vector memory to visual guidance

In the previous simulations we separately examined the effects of PI and PI-vector memory (subserved by the $h\Delta$ circuit) and memory of the current heading (subserved by the *PFN* circuit). Here we examine the consequences of combining both circuits, with associated *FBt* memory, for the visual guidance task. A single *DAN* neuron triggers the synaptic

modulation of both $FBt_\Delta$ and $FBt_{PFN}$ synapses simultaneously (Fig 5D). Therefore, any detection of the target induces both the formation of a synaptic memory of the location ($h\Delta$ pathway) and of the heading direction ($PFN$ pathway) at the last target sight. To ensure the location memory inherited from $h\Delta$ is reliable, we set the learning parameter $\beta_{h\Delta}$ to its optimal value ($\beta_{h\Delta}xI^{FBt} = 1.059$, see Fig 6C), while $\beta_{PFN}$ is randomly assigned in the range [0.0 4.0] as previously.

Fig 8A shows 4 examples of simulated paths. The paths observed with this model do not strongly differ from the model without PI except, critically, it no longer proceeds in a constant direction beyond the target (Fig 7A). The reward activity rate, of similar magnitude ranging from 0.1 to 0.2, is slightly above the model without PI showing a better ability to observe the target (Fig 8B). However, the success rate climbs to 95% (Fig 8C), and the failed attempts only happened with a very low $\beta_{PFN}$ values (<0.5, Fig 8D). Failed attempt paths mostly show really high sinuosity but still a global movement toward the target (Fig 8A-*top right panel*), probably resulting from a too short vector memory formed at the $FBt_{PFN}$ level. Finally, the success rate and time is not drastically modified by the variations of the learning parameter $\beta_{PFN}$, when it is above 0.5 (Fig 8E).

To explain the improved performance after inclusion of PI memory in the model for visual guidance, we observe that the circuit effectively sums two vectors, such that the the visually-guided vector memory ($FBt_{PFN}$) formed by the reward encounter is anchored at an allocentric reference point by the PI vector memory ($FBt_{h\Delta}$) and this is maintained if there is no new observation of the target (Fig 8F). Thus, successive observations of the visual target (or of any cue characterising the goal) trigger the creation of spatial representations of successive aims/ checkpoints around which the centre of the search pattern generated by the PI is moved. This generates more stable navigation that converges on the direct route between the last observation of the target and the target itself, whereas without the PI anchor, the visually-guided vector memory alone could quickly diverge without repeated observation of the target. Additionally, this stable spatial on-route anchor generated by the PI increases the chance to observe again the cue(s) associated with the goal.

**Effect of target disappearance.** To verify the formation of a spatial representation of the aim by the PI and visual guidance integration, we ran simulations in which the target disappears when the agent gets within a certain distance of it(Fig 9A–9C). Therefore, the behaviour of the model in the absence of any new visual information to the CX (Fig 9C) will give us insight into the emergent spatial properties in the modeled CX circuit. We tested this scenario with both the model with visual guidance circuit alone ($h\Delta$-PI pathway shutdown, Fig 9D–9G) and the model combining the PI memory (Fig 9H–9K).

The visual guidance pathway alone ($PFN - PFL$) keeps track of the agent's orientation at the last sight of of the target (Fig 9D). When it disappears, the agent is unable to stop. Combined with the motor noise in the simulation ($\sigma_{noise} = 10°$), the probability to reach the target is decreased to around 40% (Fig 9E), compared with the baseline of around 75% (Fig 7C). The decrease of the success rate seems independent of the value of $\beta_{PFN}$ (Fig 9F).

In comparison, the addition of the PI vector memory to the model allows the agent to initiate search behaviour in the vicinity of the target even when it has disappeared (Fig 9H). It preserves the high overall success rate (Fig 9I), over a wide range of values of $\beta_{PFN}$(Fig 9J). Finally, when comparing the mean distance to the target/last sight, after this last sight, we observe a strong dependence to the $\beta_{PFN}$ value. That is due to the behaviour converging to the end of the summed $FBt_{h\Delta}$ vector and $FBt_{PFN}$ vector (Fig 8F.b), whose length depends upon $\beta_{PFN}$. These results support the creation of a spatial representation of the navigation goal in the CX based on the allocentric positioning system generated by the addition of PI.

## Using a mushroom body model of visual long-term memory to navigate

In addition to navigating towards a conspicuous beacon, insects can also orient themselves based on visual memory [66, 67], aligning themselves in a particular direction with respect to surrounding landmarks or the panorama. This ability, which has been particularly associated with following familiar routes under conditions when PI information is removed, has been linked to the MB [68, 69]. The MB network, known for olfactory discrimination, can be used to discriminate on-route memorised views from off-route views [70]. However, as implemented, the MB model appear to encode valence of the visual scenery (familiar vs unfamiliar), and spatial information is encoded only implicitly in the use of retinotopic memories, such that familiarity is highest when aligned in the same direction as when the memory was stored. The quick degradation of valence with translation and rotation around a learned position [71] makes it challenging to derive robust navigation from MB output alone. If motor noise moves an agent just slightly off the route, the MB provides no useful information and it is lost. However, MB output neurons (MBONs) have been identified as strong partners for *FBt* neurons projecting into the CX in *Drosophila* [60], for example in the control of olfactory wind-guided behaviour [18]. We proposed previously a theoretical layout using the CX steering circuit to integrate the MB output, discriminate learned views oriented toward a feeder, and generate a robust navigation to this feeder [26] which is supported by the impact of CX lesions on learned visual navigation in ants [72].

Because the model we implemented in the current paper works very well with intermittent observation of the target (Fig 9), we tested its potential to integrate MB valence output. We implemented a similar MB circuit (Fig 10A) to the one we previously modeled [26]. Images from a simulated world cluttered with geometric objects are passed through an insect eye model, and lateral inhibition between blue channel ommatidia used to detect a skyline. This activates 1000 visual projection neurons (vPNs) which form random connections with the Kenyon Cells (*KC*, $n_{KC}$ = 20000) with a rate of 2–5 vPNs per *KC*. A sparse, binary activation pattern in the KCs is produced using a moving threshold to maintain the number of active KCs to 200 (1%), mimicking the inhibitory feedback regulating the *KC* activity found in numerous species [73]. Each of these *KCs* connects to a single *MBON* with an initial synaptic weight of 1. Finally, a single *DAN* neuron controls the learning process by stimulating neuromodulation at the level of every *KC* − *MBON* synapse.

The MB model is trained on a straight route (starting at the origin [0,0], randomly oriented and of 100*l.u.* length). Memories are stored at regularly spaced locations (15 locations; every 6.7*l.u.*) by activating the $DAN_{MB}$ which casuses the decrease to 0 of the *KC* − *MBON* synaptic weight of any currently active *KC*. After this training process, the agent is replaced near the origin of the learned route to observe its ability to detect and follow it. The model generates an *MBON* activity rate inversely proportional to the visual familiarity. This rate is normalised, using the fact that the active proportion of KCs is maintained constant, and binarized using an arbitrary threshold ($T_{MBON}$ = 0.01). To evaluate the performance of the MB model itself, we estimated a MB performance index, based on the alignment between the orientation of the agent when *MBON* was inactive (recognising on-route views) and the orientation at the nearest location on the route, and determined a threshold score above which the MB is considered to work better than random (i.e. appears to be more inactive when near the correct orientation) based on our data (Fig H in S1 Text). Note that we eliminated simulations with a score lower than this threshold as well as simulations where the *MBON* are never inactivated (meaning no views were recognised as on-route) from route following analysis.

The binarized *MBON* output produces a switching signal which can be used by the existing CX circuit. The *MBON* input to the CX *DAN* is set to be inhibitory, only releasing its activity

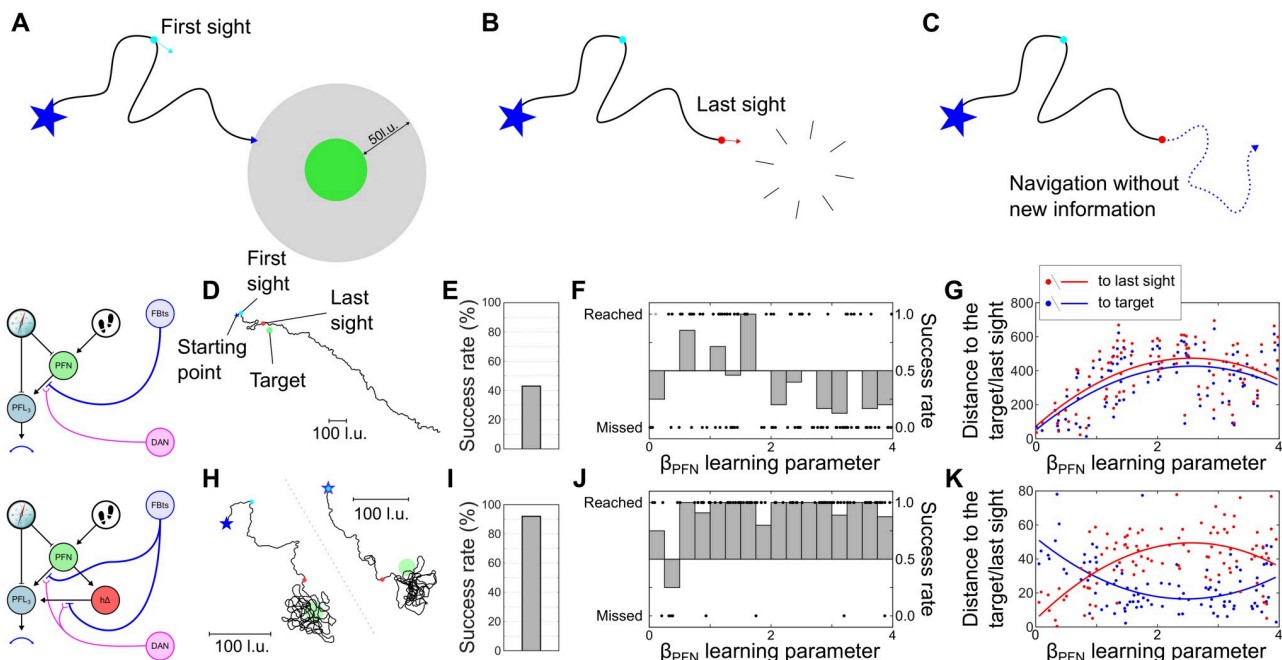

**Fig 9. Navigating when a target disappears. (A-C)** Simulation paradigm diagram. (A) The first phase corresponds to the previous simulation using visual-driven processing. (B) However, when the agent reach 50l.u. from the target, it disappears. (C) This agent is then left navigating without any new visual information modifying the *FBt − PFN/hΔ* guidance system in the CX. **(D-G)** Model without PI. (D) Example of a simulation path (black line). The location of the first sight of the target is indicated by the cyan dot and the last sight by a red dot. The target location is shown as a green plain circle and the origin location by a blue star. (E) Success rate (%) to reach the target location. (F) Success and failed attempts as a function of the $β_{PFN}$ learning parameter value. Success rate are calculated for every 0.25 section. (G) Mean distance to the target (blue dots) and the last sight (red dots) locations after the last sight event occurrence as a function of the $β_{PFN}$ learning parameter value. The lines show a 2nd degree polynomial fit for both location distances (blue for the target, red for the last sight). **(H-K)** Model with PI. Respectively identical to (D-G). *Clipart(s) in the figure have been modified from* https://openclipart.org/.

when an on-route view is detected (reward). The CX model remains identical to the previous version, generating memories in *FBt* whenever the CX *DAN* is activated. As before we compare the model with the *PFN* pathway only (memory of direction) to a model with a combined memory of direction from the *PFN* pathway and PI location from the *hΔ* pathway.

The MB model performance index shows the route views are being detected with a score above the random threshold (see Fig H in S1 Text for the definition of this threshold) in both versions of the model. However, the MB model index ranges from -1 to 1 in simulation with the model without PI, while it is tightly constrained close to 1 in simulations with the model with PI. The integration of the MB model in the CX without PI (using the *PFN* pathway only) managed to globally orient in the region of the route orientation but failed to achieve properly the following of the learned straight route all the way to the end (Fig 10C). The combined heat-maps of 6 simulations (2D histogram corrected by the orientation of each learned route) only shows a weak tendency to converge to the beginning only of the learned route (Fig 10C, *Right panel*). In comparison, the addition of the PI (*hΔ* pathway) and the associated vector memory circuit successfully generated navigation behaviour converging on the learned route. Moreover, the combined heatmap of all simulations indicates a strong peak around the end of the learned route (Fig 10D). The PI model, compared with the model without PI, shows better performance in the average distance to the route, in the extent of the route followed (see Fig G in S1 Text), and the minimal distance to the end of the route (Fig 10E). Note that the PI model's average distance to the route is kept low mostly by its ability to restrain the navigation in the

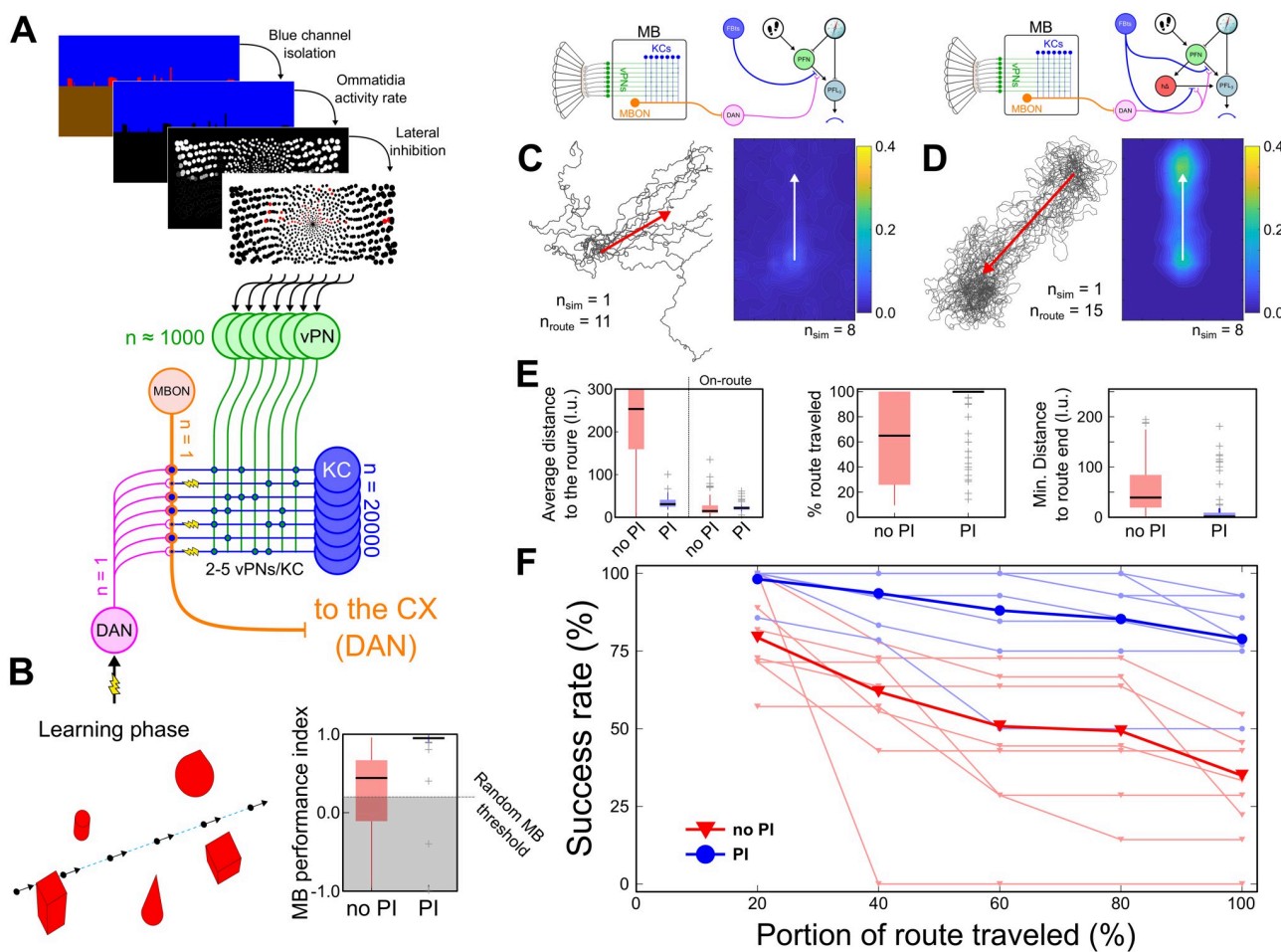

**Fig 10. Navigating using the MB long-term visual memory. (A)** Mushroom body model. Visual input is based on the blue channel of an insect-inspired eye model (see Fig B in S1 Text), with lateral inhibition to encode edges in the layer of visual projection neurons (vPN) that provide input to the MB. **(B)** *Left panel* Diagram of the initiation of the MB "snapshot" memory following a straight route. *Right panel* Boxplot of the MB model performance index for the 2 versions of the model (see details in Fig H in S1 Text). **(C)** Simulation without PI. *Left panel* Path of 15 route following attempts for 1 single route learned. The red arrow indicates the learned route. *Right panel* Heatmap of 8 pooled simulations, realigned on the learned route direction (red arrow). **(D)** Simulation with PI, similar to (C) panels. **(E)** Boxplots of, from left to right, the average distance to the route (on the whole path and on the path detected on the route, see Fig G in S1 Text), the percentage of route traveled and the minimal distance to the end route. All the data have been filtered based on the MB performance index (B). **(F)** Success rate measured at different portions of the route for the model with (blue stars) or without (red triangles) PI. Individual simulations (1 single route learning and 15 route following attempts) are indicated in thin lines while pooled data are indicated in thick lines. *Clipart(s) in the figure have been modified from* https://openclipart.org/.

vicinity of the route after losing it, whereas the model without PI diverges quickly when it loses the route (Fig 10E). Finally, the performance of route following decreases quicker in the model without PI along the route (Fig 10F). These results further demonstrate how the spatial representation of the navigation goal is enabled by the addition of PI to the sensory guidance circuit.

In addition, we tested the MB-guidance on more complex routes. We generated zig-zag routes similar to the first outbound journey in the vector-memory paradigm. While progressing along this route, the MB long-term memory was created continuously, during each step of the model ($0.25 lu.st^{-1}$), by maintaining $DAN_{MB}$ active with a learning rate readjusted to 0.1. After the entire route has been learned, the agent was replaced around the starting location

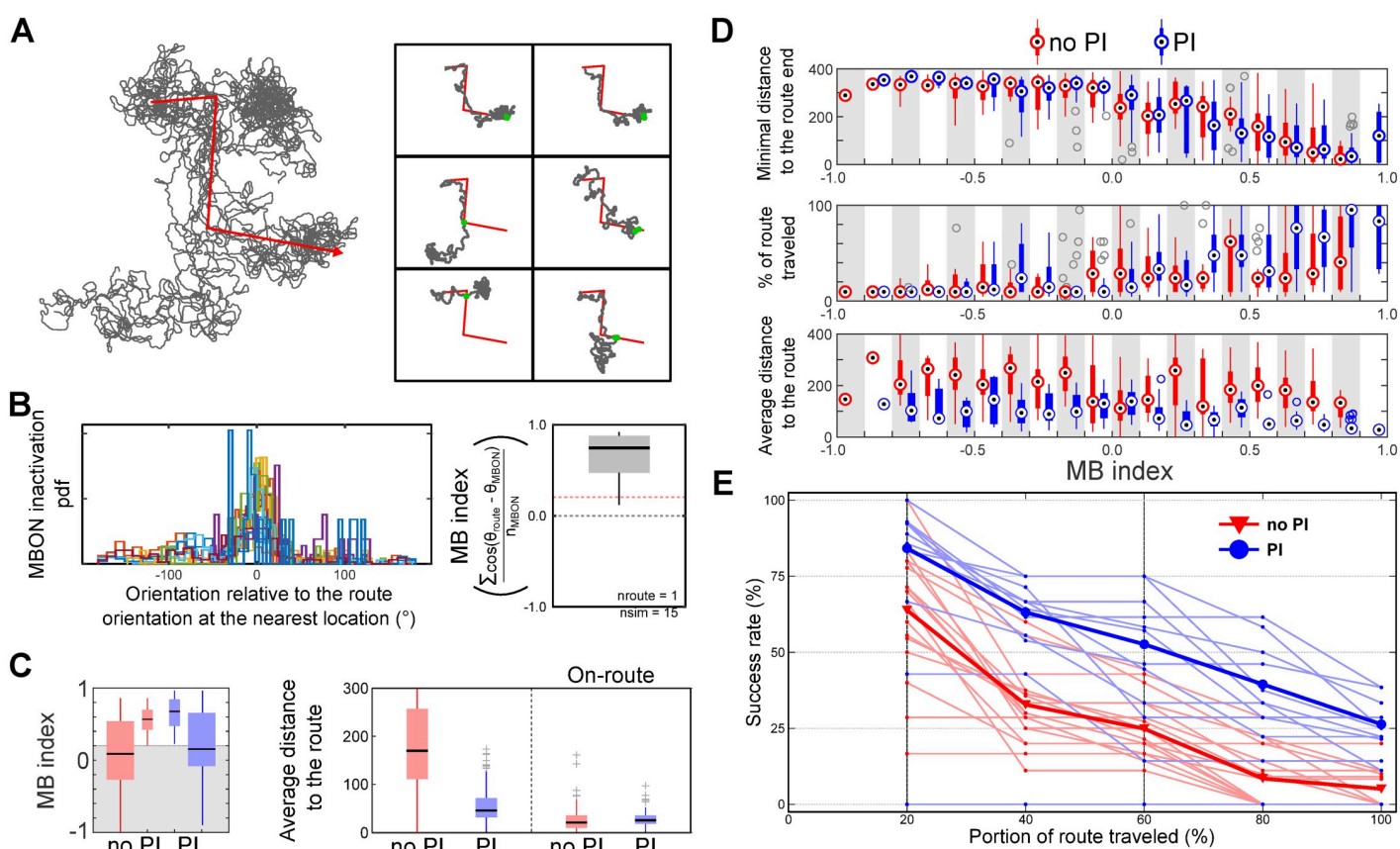

**Fig 11. Route following using the MB visual long-term memory.** (A) *Left panel* Simulation example of a single route learning paradigm (red zigzag arrow) and route following trials (black lines, n = 15). The MB learning, only during the walk (with constant speed) along this route, is continuous and modulated by a learning rate of 0.1. All route following attempts are initiated at a random location situated around the starting point of the route (±20*l.u.*). *Right panel* Individual trial of route following. The route is shown as a black line and the route following attempt in blue. The red dots indicate the position at which the route has been considered lost (see Fig G in S1 Text for calculation details). (B) Estimation of the MB model performance scores for 15 attempts to follow a single learned route *Left panel* Probability density function of the *MBON* reward inactivity (on-route views) in relation to the orientation relative to the nearest route orientation. *Right panel* Boxplot of the MB evaluation scores (see Fig H in S1 Text for calculation details). (C) *Left panel* Boxplots of the MB performance score for the two different versions of the model (no PI: red, PI: blue) before (outside large boxplots) and after (inside small boxplots) the elimination of low MB performance score (see Fig H in S1 Text). *Right panel* Boxplot of the average distance to the route with both versions of the model calculated on the whole path or only before the route was considered lost (see Fig G in S1 Text). (D) From top to bottom, boxplots (red: without PI, blue: with PI) of the minimum distance from the route end, the percentage of route traveled and the average distance to the route in relationship with the MB performance score. Each boxplot is based on the pooled data of simulations with MB performance index value within a 0.1 range (grey and white bands). (E) Success rate of the model to reach different portion of the route. Data have been selected based on the MB performance score threshold. *Clipart(s) in the figure have been modified from* https://openclipart.org/.

with a random orientation and the route following was tested. For each route learned, 15 tests were conducted, each with a different starting location and orientation.

Fig 11A shows the example of a full route following simulation, 1 route learning and 15 route following attempts, and 6 examples of individual route following attempts for the same route learned. Without additional components, the model shows capability in tortuous routes following. With both model versions, the MB performance index lies mostly under the random threshold, indicating it is a more difficult task to recognize views of combined route segments in different orientations (Fig 11C). Note that our method to define this index and the estimation of a nearest route location might add some discrepancies in the vicinity of the zigzag turns, where orientations vary quickly from one segment to another. Despite, while both models show an equivalent ability to approach the end of the route, the model with PI clearly shows an

improvement in term of the route following and the average distance to the route (Fig 11D). The performance of both model is however strongly limited by the performance of the MB itself (Fig 11D). The addition of PI clearly increases the capability of the CX model to sustain route following over more complex routes (Fig 11E). Particularly, the performance of the model to follow the route in the vicinity of turns (20% and 60% of the route) diminishes linearly with the PI whereas on average, without PI the success rate drops more rapidly after each turn.

## Discussion

We propose an integrative model that follows the functional neuroanatomy of key identified neurons of the insect CX [12, 60], from the head direction to the steering circuits. Our main guideline to build the CX model was to follow closely the known projectomic/connectomic data in insects [36, 60]. However, we focused on the common patterns of projection and their functional aspects [12], rather than on the fine and species-dependent details. From this approach we aim to extract generic insight into CX processes that could guide our understanding of its function(s) as well as its notable conservation across insect evolution.

Specifically, we suggest that identified neurons (*FBt*) that convey signals from outside the CX and project across the FB serve the purpose (previously hypothesised without explicit neural grounding in [20]), of storing vector memories. This is merged with previous models that suggest integrative processes in intrinsic FB neurons (*PFN* and/or *h*Δ) can support path integration and homing to a nest [13] and persistence in approach to targets that provide a sensory-based, innate or learned, reward [26]. The connectivity between these components enables flexible switching between navigational modes. We show the model can support a range of behaviours including: revisiting a food source location; maintained navigation towards a visual target that disappears from view; and route following based on a familiarity signal from the mushroom bodies. We argue that this circuit function of creating allocentric spatial goals (not just heading goals) for immediate navigational control is relevant to a wide range of insect species and behaviours.

To our knowledge, no direct behavioural evidence exists as yet to support such a spatial representation in insects. However, we believe the development of experimental paradigms within a cluttered environment, forcing constant obstacle avoidance, and/or controlled disappearance of the sensory cues, mimicking our implementation (Fig 9), could provide insight to the actual existence of spatial representation of navigation goals. That is, we predict that an insect, after contouring an obstacle [74, 75], should set a course that reflects its last estimate of the goal position, or a point on route to the goal, not just the goal direction; in a similar manner to how ants have been observed, after deflection, to set a direct course to the nest [76] or to a known feeder [66].

### Spatial vs. heading goals

The key insight from this integrative model is the emergence of an allocentric navigation vector—indicating a specific spatial location as the current goal to move towards—from the combination of both the directional ($FBt_{PFN}$) and positional ($FBt_{h\Delta}$) memory circuits. To date, the CX circuit has been most widely interpreted as a system that allows the insect to maintain a desired heading, supporting behaviours such as persistence in a direction when cues disappear [77], menotaxis, i.e. the ability to maintain a constant bearing in regard to a particular cue [27], or simply to walk straight [28]. More generally, in the navigation context, the CX has been highlighted as as neural circuit that compares the heading of the insect with its desired heading or goal direction [13, 26, 34, 78]. This has been supported by recent neuroimaging and behavioural data, demonstrating the mechanism that allows insects to steer based on the

difference of headings (actual and desired) [31, 52, 79]. We have previously proposed that if the goal direction is determined by a 'home vector', obtained through path integration, then such a steering system can guide the insect to a particular location in space, specifically, the place where that vector is zero in length [13]. As such, the assumed primary function is to return to a point of origin such as a nest [15], but by introducing an opposing 'vector memory' the 'zero point' can be positioned at an alternative location relative to the nest, such as a recently experienced food source [20, 80]. Importantly, this system also produces an emergent search pattern around the target location without any additional mechanism or explicit switch to search behaviour required.

Here we suggest that PI and vector memory are not just specialised functions for central place foraging, but rather play a more general role, intrinsic to how the CX supports all spatial behaviour. The high degree of conservation in CX structure across insect species and the existence of PI in species such as *Drosophila* [22, 23] support this extension. For example, a simple modification of the circuit—abolition of the return inhibitory pathway—sustains navigation and exploitation of several food sources in a 'nomadic' fashion (Fig 12), reminiscent of *Drosophila* exploratory behaviour [81]. Essentially, we have shown that PI can provide an anchor, i.e. a stable reference/origin point in space, for any sensory-guidance pathway, extending its

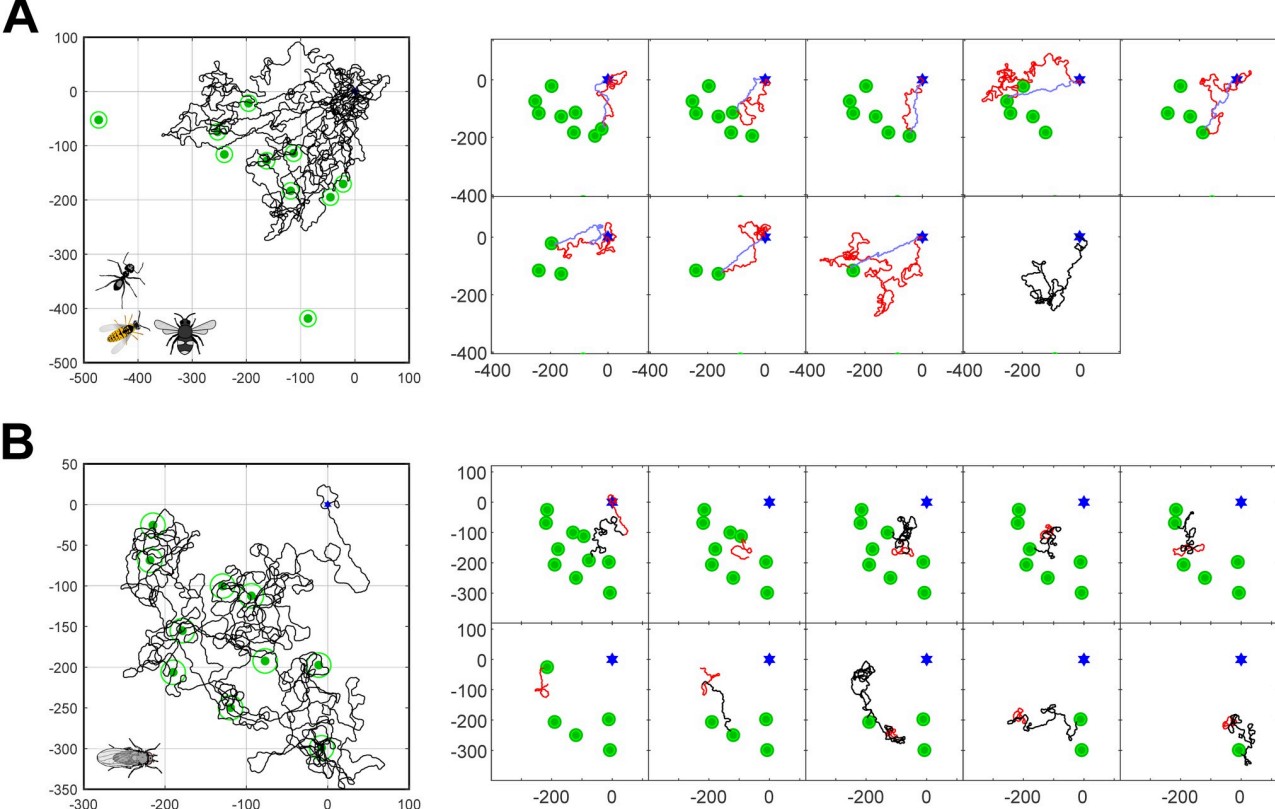

**Fig 12. Simulations using a "settler" or "nomadic" CX to harvest multiple food sources.** (A) *Settler brain* The CX model used in this simulation is similar to Fig 5B.c. Both sensory stimulations from the green visual pathway and food discovery are used as reward to the *FBt − DAN* circuit. The agent is placed at the nest and left to find food located at different location randomly assigned in a quadrant of 90˚ and at a distance of 100–400 l.u.. When a food source is found, it is fully consumed and disappears. Simultaneously, a memory is formed on the *FBt*^hΔ and the motivation is switched to trigger homing behaviour. When the agent reaches the nest, the motivation is switched again to promote further exploration. The simulation is stopped when the agent spends more than 5000 t.u. without finding a new food source or the nest and is considered lost. (B) *Nomad brain* The CX model used in this simulation is modified to replace any *DAN* stimulation by a re-zeroing of the *h*Δ activity, i.e. of the PI. The rest of the simulation is similar to (A). *Clipart(s) in the figure have been modified from* https://openclipart.org/.

function from self-positioning to goal-positioning (we note that [82] similarly suggested that the fly may compare its vectorially encoded position to a vectorial goal, in 2D or 3D space, although not presenting a specific mechanism). In the example behaviours we have simulated, this 'goal' does not correspond to the exact location of the ultimate goal (such as the green cylinder target or the end of the learned route). Rather, the location temporarily imprinted in the memory is a way-point on route to this ultimate goal. This constrains movement to stay close to this route, by integration of deviation from the desired way-point, facilitating the (re-)observation of either the goal or any cue associated to it. Therefore, the behaviour innately converges, thanks to the innate PI central displacement, to a location that get closer and closer to the targeted location, maintaining a robust balance between exploration and sensory-guidance. Animals in the wild are confronted with cluttered environments and must deal with sparse sensory inputs and/or detours [75, 83]. Being able to keep a trace of a location (where you were headed when the goal was last seen) rather than just a direction (which compass direction you were headed when the goal was last seen) must therefore represent a significant advantage. For example, as we have shown, it can significantly improve a route following by an agent using visual familiarity learned in the MB [70] which otherwise can get quickly lost when small displacements or rotations make all views appear unfamiliar [71], and cannot recover the route using the heading memory alone. Other recent proposals to link the MB visual long-term memory to CX steering have used the bilateral MB to learn views corresponding to left and right turns in a visual homing task [78]. This extension would be compatible with our implementation and could further improve the ability of the model to follow tightly a route.

The association of visual memories and PI, either to follow a route or to achieve homing, has been previously modelled as a problem of reconciling independent pathways [34], each one carrying a goal direction to follow. This approach is influenced by the description of insect navigation as a toolkit containing parallel functions [32] that should be optimally combined at output [7, 35]. By contrast we propose here a more integrated architecture, where different pathways contribute with complementary properties. Therefore, the outcome of the CX processing, illustrated here by the visual-driven and the route following behaviour, benefits simultaneously from the directional inputs of the compass, from the positional inputs of the PI and from the visual memory of the MB to achieve a more robust navigation than previous implementations [26, 70]. In addition to improving visual memory navigation, the same principles could also benefit other pathways connected to numerous *FBts* [60]. Note, however, that this integrated circuit does not explicitly associate a PI (location) to specific visual memories, as would be required to form a cognitive map [7]. Here, the spatial memory is created locally and only serves to navigate using the immediate sensory inputs. However, the storage of locations in the CX based on different sensory inputs could help explain cognitive-like ability observed in insect navigation behaviour. Additionally, the transformation of sensory/memory inputs to the CX from egocentric directional to allocentric positional representations could be crucial to unify modalities often presenting different spatio-temporal dynamics, allowing insects to more robustly navigate in their environment.

## Comparison to alternative models

A number of other CX models have been proposed that utilise essentially the same neuron classes and connectivity as our model, but with somewhat different functional interpretations. Here we summarise the key similarities and differences, and relevant experimental evidence supporting each.

**PFN—functional compass copy.** *PFNs*, in our implementation, serve to create a functional copy of the compass bump carried by Δ7, with their projection pattern producing a

bilateral 45˚ shift of the bump to the left or right. In principle, the inhibitory Δ7 synapses are modulating, across the FB, an activation rate of the *PFNs* which is determined by the agent's translational motion. In the current simulations, the speed is always set to a constant value, and we neglect possible holonomic motion, hence the baseline activation is always equal in left and right *PFNs*, and the activation pattern is simply the inverse of the Δ7 pattern, corresponding to a sinusoid-like function with a phase centered on the *EPG* bump of activity. In a previous model [13] the left and right *PFNs* are activated from neurons in the noduli tuned to optic flow at ±45˚ forming an orthogonal basis that allows egocentric translational inputs to be transformed to an allocentric estimate of motion that is accumulated to encode PI memory. More recent models, supported by neurophysiological measurements in flies, suggest that different subtypes, $PFN_v$ and $PFN_d$, between them form a full orthogonal basis for the translational motion of the insect in allocentric coordinates, which is integrated by hΔB cells into an allocentric vector of motion [16, 17], see also [60]. These studies do not find evidence for accumulation of the translational input in $PFN_v$ and $PFN_d$; although the existence, in bumblebees, of a pathway in the cap region of the noduli targeting specific PFNc neurons [36] that are not present in flies, leaves the possibility open that *PFN* could support PI in some insect species. In [84], $PFN_a$ are shown to be sensitive to wind from ±45˚, and in the model presented in [18] this is assumed to support an analogous allocentric estimate of the wind direction in hΔA cells, which can contribute to upwind steering (as described further below). In brief, the interpretation of *PFN* neurons as not just copying and shifting the compass signal, but as transforming egocentric self-motion into an allocentric vector remains consistent with our model's assumption that this vector provides a (temporary) goal ("maintain recent heading"). However we note that introducing holonomic motion in our model would require a compensatory mechanism (similar to that included in the model in [13]) to deal with the right-left imbalance in *PFN* activation this would produce, which would otherwise bias the memory processes (PI and vector memory) and the steering control.

**hΔ—180˚ phase-shift and integration.**    hΔ are essential to our model. They allow a 180˚ rotation of the compass and, by integrating the *PFN* inputs over time, form the PI homing vector, which then becomes a potential goal for the steering circuit. Previously, in [13], where it was assumed that *PFN* accumulated the PI vector, the analog of hΔ in bees (the pontine neurons) were only used to normalise the bilateral signal from *PFN* reaching the analog of *PFL* (CPU1). As indicated above, recent models in *Drosophila* suggest that by unifying the $PFN_v$ and $PFN_d$ in the same reference frame, hΔb allow the transformation of the heading orientation, acquired from external cues (compass), into an allocentric travelled orientation, based on the forward-backward and the 45˚ shifted optic-flow velocity estimation on both hemisphere [16, 17, 60]. This property remains consistent with our model, as this is indeed the velocity vector that should be accumulated for PI. Interestingly, [60] suggest that recurrent connections between hΔ pairs might support PI, but as yet there is no direct evidence for this. We note that using hΔ instead of *PFN* for PI allows the model to represent simultaneously the "recent heading" and "home" directions, whereas using *PFN* as the substrate (as in [13]) only allows "home" and "anti-home" directions. In the model proposed in [18], where hΔc are assumed to represent allocentric wind direction, an additional functional role of these cells is to integrate contextual control signals for the presence of odour, in which *FBt* neurons gate the hΔc output to *PFL* to drive up-wind steering.

**FBt—Dynamic goal direction.**    The key novel component of our implementation to generate a goal direction signal, and the most speculative, is the tandem *FBt − DAN*. While based on the neuroanatomy of the *FBt* in *Drosophila* [60], the exact connectivity of these cells remains speculative. The addition of the *FBt − DANs* updates our previous model, in which such tangential FB inputs only conveyed a reward signal from innate or learned visual

pathways [26]. This signal modulated integration in the FB to create persistence in rewarded heading directions and it provided an anatomically grounding for the theoretical implementation of vector memory in [20]. Note here that in addition to axonal connections, used in our model to stably maintain the vector memory, *FBts* present projection to the dendritic region of *h*Δ subtypes [60] that are not considered in our model, due to the single linear unit neuron model. It remains to show what function(s) they could serve but we can speculate that dendritic projection would interfere with the inner computation of the neuron, and particularly the PI in the case of *h*Δ, as we hypothesise here. Therefore, introducing them in the model could be useful to set/reset the PI at a specific state (based on sensory/memory information, zeroing at the feeder in drosophila for example [25]) or eventually gating it (when the orientation/odometry become uncertain for example). In the current model we extend the function of the vector memory circuit to form a goal direction signal. Similarly, the neuromodulation of synaptic weights to store the goal heading in the FB has been also proposed to sustain a contextual control of saccade-fixation *Drosophila* behaviours in a negative reinforcement paradigm [85]. In all cases, the resulting goal direction is represented by a sinusoidal pattern in the FB, which allows its effective comparison with the equivalently shaped compass signal in the PB [12, 52]. Indeed, a goal representation in line with these theoretical predictions has been experimentally verified in *Drosophila* [31]. In those experiments the animal's goal direction could be optogenetically controlled by the activation of FC2 neurons, columnar neurons with mixed input and output fibers in individual columns of the FB [31]. This concept of a goal direction population code differs from our implementation of a goal direction encoded in *FBt* output synaptic weights. However, it is still unknown if the sinusoidal FC2 activity pattern is generated within these cells or inherited from upstream neurons. As the intra-columnar projections of FC2 cells suggest local information flow within columns of the FB, it is conceivable that the goal representation does not originate in FC2 cells, but that these cells could carry the associated information from where it is stored to where it is used; i.e., in our implementation, from *FBt* output synapses to the location where *PFN* and *h*Δ neurons interface with *PFL* cells.

To date, the only other models to include *FBt* neurons address olfactory navigation behaviour in *Drosophila* [18, 86]. The existence of both olfactory and visual pathway in parallel influencing the CX navigation is supported by multiple *FBts* input to the FB in insect [60], potentially representing different sensory-based vectors, supporting the multimodal control of navigation [87]. The main projection pattern of *FBt* they propose is similar to ours and their upstream input—non-directional odour perception/recognition, potentially from the LH (innate) and the MB (learned)—can be interpreted as a reward input, i.e. matching the dopaminergic *FBts* (*DAN*) in our model. They propose that *FBt* gates the columnar neuron (specifically *h*Δ*c*) outputs to generate different goal directions based on the olfactory context, resulting in upwind orientation when exposed to attractive odours. While similar in principle, this model differs in two ways from our new model. Firstly, it does not use PI, and second, there is no dynamical modulation of the goal direction. The latter means that the this model is limited to the two predefined alternatives of upwind and downwind flight direction. Although this binary choice is sufficient for wind guided olfactory navigation when locating the source of an odor plume [88, 89], recent observations of continuous adjustment of goal direction signals in monarch butterflies after negative conditioning [79] support the capacity of the CX to perform dynamic readjustment of the goal direction based on sensory experiences—consistent with our implementation. Interestingly, in the event of a loss of the odour plume, insects show a stereotypical casting behaviour, sweeping from right to left in quick alternations to recover the odor plume [90]. We suggest this could correspond to the searching behaviour generated in our model, which is induced by the PI at the location indicated by the combination of the most recent vector memories (homing and sensory vectors). In the context of odour plume

following this should correspond to the last place estimated on the route to the plume source, thus increasing the chance that the odour molecule can be detected again.

In summary, tangential inputs to the FB (*FBt* neurons) represent an efficient means to map non-direction inputs onto the directional system formed by columnar neurons [18, 26]. Our model follows this principle and additionally introduces positional inputs that strengthen the sensory-guided navigation by complementing it with the path integration at the same level. The synaptic modulation to represent dynamical changes in the goal direction memory is speculative, although consistent with the adaptability [79] and the persistence [77] of insect navigation behaviour. Additionally, such synaptic modulation is supported by the evidence for dopaminergic circuits in the FB [60], and the well-known function of dopamine in plasticity and learning, for example in the MB memory [91] or in the ER neurons plasticity [47]. Therefore we think our model provides a reasonable prediction of *FBt* function and could inform future experiments on the interplay between *FBts* and the FB pool of columnar neurons (*PFN*, *h*Δ [18] or FC [31]. However we note that the mechanism that sets the synaptic weights in the model is very abstracted, and lacks any obvious foundation in biophysical mechanisms of plasticity; this should be a focus for future work.

**PFL—CX steering output.** The unilateral signals carried by the different component of the model (Δ7, *PFN* and *h*Δ) are then summed and compared, bilaterally, at the level of the *PFL* which compute the steering output of the CX model, as in previous models [13, 20]. Their role is supported by evidence of a strong downstream connectivity of $PFL_3$ to descending neurons (DNa02) responsible for turning behaviour [51, 60], and has been recently verified by functional neuroimaging and modeling [31, 52]. However, the connectivity pattern we used, presenting a strict separation between hemispheres and offset by exactly one column, does not follow the precise details of the projection of either $PFL_1$ and $PFL_3$ reported in *Drosophila* [60], which have (respectively) a one ($PFL_1$) or two ($PFL_3$) column offset pattern that continues across the midline. All these patterns can reproduce the key function of creating a left-right difference in the *PFL* that reflects the difference between the compass and goal directions (Fig F in S1 Text), but incorporating the real connectivity in the future might allow a more subtle control of the steering. While it is generally accepted that relative $PFL_3$ activity affects turning direction in the animal, different models have used this signal directly to set angular velocity (our model, [13, 18, 34, 52]), or to influence the relative probability of turning in each direction [85], or as input to an intrinsic oscillator based on the LAL circuit [92, 93]. The role of $PFL_1$ could be simply to refine the $PFL_3$ signal, but has also been speculated to instead contribute to head motion, or to controlling directional change through side-slip instead of body rotation [60]. Our model used a constant forward speed and so we did not include $PFL_2$ neurons, which have a four column offset and thus combine information from both sides of the FB; in other models they are assumed to modulate forward speed or fixation duration in *Drosophila*. $PFL_2$ activity has also been suggested to increase the angular velocity when facing the anti-goal [52], as the difference in $PFL_3$ activity has a minimum at this point. However, we note that facing the anti-goal (unlike facing the goal) is an unstable state in the CX steering system. Hence the motor noise of ($\sigma_\epsilon = 10°$) used in our agent simulation (or alternatively, intrinsic oscillatory behaviour [94]) can turn the agent away from the exact anti-goal direction sufficiently that differential $PFL_3$ activity will then continue to turn it back towards the goal direction.

## Potential for temporal integration to improve the goal estimate

One limitation in the current implementation is the absence of any dynamics in the vector memory (goal) formation at the level of the *FBt* output synapses. The navigation is therefore based on a vector of arbitrary length (depending on $\beta_{PFN}$) and completely rewritten with every

reward event. A more realistic neuromodulation dynamic could induce a vector formation process that averages over events, and therefore space, to refine the positional goal gradually as more information is acquired. We showed in a previous implementation of the MB to CX pathway that the integration of the MB output signal over time could be used in this way to strengthen the navigation toward a learned feeder [26]. Additionally, observations in *Drosophila* are consistent with the assumption that they continuously adjust the zero-point of their PI to the statistical center of a optogenetically rewarded area [25]. Adding such a mechanism to the model in this paper could support a form of triangulation, relying on the positioning ability of the PI, that continuously improves the estimation of the navigation goal location every time an attractive/rewarding sensory combination is experienced.

## Contextual control of navigation in insects

The recent deciphering of the FB circuit and its implementation in functional models [18, 31, 52] has strengthened the hypothesis that its key role is the comparison of the goal and current heading directions. The existence of a wide variety of converging sensory streams through a pool of specific *FBts* [60], should thus support multisensory integration, matching behavioural evidence of multisensory navigation in insects [40, 87]. Moreover, the mechanism we have highlighted, inducing a heading vector memory under the control of a contextual reward, is perfectly suited to allow the selection of action based on several vectors, either learned or based on immediate sensory inputs, in line with contextual gating in wind-guided behaviour in *Drosophila* [18]. Therefore, in addition to the integration of different goal orientations together [19], the FB could be involved in a dynamical selection of the different modality(ies) depending on the immediate information/context available. More specifically, our model's ability to not only inherit a goal orientation from external sensory streams but rather to rebuild it plastically based on the inner compass orientation and on contextual valence signals, supports several sensory combination mechanisms, such as association, gating and/or summation [95]. The layering of the pool of *FBts* existing in insects [60] is particularly adapted to this selective activation/inhibition as well as the dynamical construction of the different sensory-based navigation vectors. Consequently, the FB appears as a structure combining a set of vectors, sensory- and/or memory-based, expressed in a common reference frame to achieve a context-dependant navigation task.

## Analogy to vertebrate navigation

As discussed above, our model suggests that directional information from egocentric sensory cues and memory is transferred into an allocentric spatial framework. Importantly, this framework does not require the formation of a cognitive map to support successful navigation. Whereas the existence of a cognitive map in vertebrates has been supported by the discovery of a variety of spatially tuned cells [1, 2], an equivalent construct in insects remains questioned [5, 6]. So far, cellular evidence comparable to that in the vertebrate hippocampus and entorhinal cortex is absent in insects. However, a map-like representation that could be supported by the insect brain's more moderate coding capacity might rely on an entirely different set of cell types. In principle, a map-like representation would imply that memories of sensory experiences in particular locations are associated to the geometric coordinates underpinned by PI [33, 96, 97]. This ability would allow an animal to recover PI vectors based on sensory and memory inputs, as well as to predict the sensory inputs based on the PI state [7]. Clearly, the model circuit we explored in this study is not sufficient to support a cognitive map representation. While it generates a long-term memory of a location based on the PI vector [20], it can neither recall these vectors based on memorized sensory information nor recall sensory

memories (MB stored "views") based on the PI state. Nevertheless, the proposed *FBt* vector memories in our model are based on both PI (*FBt − hΔ*) and innate and/or memorized sensory cues (*FBt − PFN*). They therefore could constitute a key circuit element to achieve bidirectional cross-talk between PI vector memories and memorized sensory cues. A circuit supporting a cognitive map should also involve neurons projecting from the CX back to upstream sensory and memory regions, for example FS neurons, columnar outputs from the FB identified in *Drosophila* [60], which project to several brain areas including the MB [98]. This prediction provides a unique starting point towards interrogating the feasibility of a cognitive map in the insect brain.

## Supporting information

**S1 Text. Supporting methods and figures. Fig A. Virtual worlds used in simulations.** Concentric circles on the ground do not appear during the simulations and are displayed here to show distance, 100l.u. separate consecutive circle radius. Vector memory replication of Le Moël et al [20] have been conducted in the empty environment. The sensory attraction to the innate green cylinder have been conducted in the single landmark. The MB route following simulation, both with a straight or zigzag route, have been conducted in both the enriched environment and the cluttered environment, without green landmarks in it. Finally, multiple source exploration simulations have been conducted in the cluttered environment, with randomly positioned green landmarks as food sources. **Fig B. Eye model visual processing.** We built an eye model with the aims to **(A)** reduce the resolution inherited from the raw simulation images and **(B)** represent the heterogeneity observed generally in insects eyes [99–101], i.e. the frontal and horizon part presenting often a higher resolution than the rest of the eye. **(C)** Each ommatidia is then assigned the set of pixel corresponding to its projection on the pyOpenGL rendering planes (4 orthogonally organized planes forming the panoramic view with a 160˚ vertical span. The activity rate of an ommatidia is calculated by the averaged light-level of all its assigned pixels. **(D)** The two color channels visible by insects (Green & Blue) are separated and can be used for different pathways. The Green channel is defined to create the frontal (above the horizon) visual field detection of green landmark (innate). Alternatively, the blue channel is used to input into the MB through the vPNs (learned). **Fig C. CX model connectivity matrix. (A)** Generic connectivity pattern used between individual neuron group. The ID of each neuron of a single type is based on the exitence of the functional columns observed in several CX sub-structure. **(B)** Overall connectivity matrix representing the whole CX model. **Fig D. Compass circuit.** The compass circuit consists of a ring attractor distributed between the EB and the PB that has been highlighted in recent neurophysiological studies in insects [9, 102, 103]. **(A)** Compass circuit diagram. The circuit is represented in a circular fashion to shed light on the columnar organization across EB and PB. The inter-neuron connectivity pattern is only shown for one functional column and repeated identically for every other. Note the intrinsic connectivity pattern across the PB of Δ7 **(B)** Compass orientation input to the *EPG* (EB). The orientation of the agent is compared with the preferred directions (with a 45˚ acceptance angle) of 8 orientation sensitive cells (Compass Neurons, CN), which could therefore correspond to mimic a sky polarization pathway. The cell that is sensitive to the current orientation is set with an activity rate of 1 while the others activity is set at 0. Each of this CN synapse to both *EPGs* of each wedge, one for each hemisphere, in the EB. **(C)** Compass rotational input to the *PEN* (PB). Left/Right turns alternatively excite *PEN* on one hemisphere of the PB allowing the rotation of the compass according to the movement of the insect. **(D)** Compass reformatting from the sensory input to the Δ7 layer and distribution to the *PFN* layer. Through the different layer of the compass circuit and particularly due to the Δ7

projection pattern, the single activity 'bump' inherited from the CN is transformed into a sinusoidal shape signal inherited by the *PFNs*. **Fig E. Model with sinusoid compass inputs (A)** Head direction signal process from the Compass Units, following a winner take all activity rule, to the *PFNs*. **(B)** Head direction signal process from the Compass Units, following a sinusoidal activity rule, to the *PFNs*. The activity of each units is calculated as the sinusoid of the difference between the orientation of the agent and of each compass unit preferred direction. $k_{\Delta 7}^{EPG}$ is adjusted to 0.4 to deal with the increase in overall activity across *EPGs*. **(C)** Vector memory paradigm (Fig 4A) using the sinusoidal compass input. **Fig F. Model with a realistic $\Delta 7 -$ $to - PFL_3$ connectivity pattern [31, 52]. (A)** Default connectivity used in our model. No shift is applied from $\Delta 7$ to *PFL* as we used previously in [26]. **(B)** Connectivity pattern following a 2 columns shift from $\Delta 7$ to $PFL_3$ as observed in *Drosophila* [31, 52, 60]. Note that connectivity pattern are presented without the additional 4 columns shift inherited from the $\Delta 7$ projection pattern across the PB for simplicity. **Fig G. Route following detection routine. (A)** Checkpoints are homogeneously distributed along the learned route. **(B)** For each route following trial, we calculated a minimal distance to the checkpoints, each line indicate a route following attempt (n = 15). Trials where the route is lost show typical escape line, which indicate a linear increase of the minimal distance to the checkpoints along the route whereas route following behaviour should be characterize by a constant (and lower) minimal distance to checkpoints. To identify this linear increase of the minimal distance to checkpoint we calculated its rate along checkpoints. We then defined a threshold based on the distance inter-checkpoints ($T_{rf} = 0.75 D^{ck}$ with $T_{rf}$ the threshold and $D^{ck}$ the distance between two consecutive checkpoints). Whenever the minimal distance variation between two consecutive checkpoint failed to stay under this threshold ($Min(dCk^i) - Min(dCk^j) > T_{rf}$; with $dCk$ the distance to a checkpoint $j = i + 1$), it indicates the agent never get closer than 0.75 times the distance inter-checkpoint than it was at the time it reach the minimal distance to the previous checkpoint, showing a lack of progress along the route. The checkpoint is then tagged as not visited and two consecutive checkpoints not visited are considered an end to the route following behaviour. The last checkpoint visited is therefore the further on-route location. Note that we did not consider any return to route following after it was consider out once. **(C)** Example of the route following end point detection by the routine described previously on 15 trials of a simulation. Red dots indicate both the further checkpoint reached on the learned route (green line) and the closest location to this checkpoint on the retrieval attempt (black line). **(D)** Boxplot of the percentage of route traveled based on the estimation of the last on-route checkpoint visited. **(E)** Success rate (%) at different portion of the route. Note that the learned route turns always happen at 20% and 60% of the "zigzag" routes. **Fig H. Calculation of the MB model performance index. (A)** To estimate the intrinsic performance of the MB model in route following simulations, we define a performance index based on the alignment of the agent at time where the *MBON* is active (views recognized as on the route) and the actual orientation of the nearest route location. **(B)** Alignment of these orientation indicate a good recognition and the cosinus of the orientation difference lie close to 1, whereas perpendicular orientations lie around 0 and

opposed alignement around -1. The averaging of all these comparison ($\frac{\sum cos(\theta_{MBON} - \theta_{route})}{n_{MBON}}$) therefore range from -1 for an overall anti-alignment between the actual route and the recognized route, to 1 for a perfect alignment/recognition. **(C)** For each simulation we therefore estimate a score indicating the actual intrinsic performance of the MB model. **(D)** To estimate a limit/threshold indicating that the MB model was better than random we benefit from simulations where the MB was actually constantly active, indicating an unspecific recognition of on-route views. We then arbitrarily define a value just over the range of MB performance indexes estimated for these simulations to define our threshold. All simulations with a score lower are

considered impaired by the MB model itself and excluded from analysis. **Table A. List of neuron-to-neuron gain parameters used in simulations** ($K_{output}^{input}$) Note that parameters are not set to reflect a biological reality but rather to ensure a stable function of the model. However, no automatic optimization process have been applied to define this particular set of parameters. (PDF)

## Acknowledgments

We are very thankful to people from both the Insect Robotics group in Edinburgh and the Vision group in Lund for the fruitful discussion that helped improve the development and the description of the model.

## Author Contributions

**Conceptualization:** Roman Goulard.

**Data curation:** Roman Goulard.

**Formal analysis:** Roman Goulard.

**Methodology:** Roman Goulard.

**Supervision:** Stanley Heinze, Barbara Webb.

**Validation:** Stanley Heinze, Barbara Webb.

**Visualization:** Roman Goulard.

**Writing – original draft:** Roman Goulard.

**Writing – review & editing:** Roman Goulard, Stanley Heinze, Barbara Webb.

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
