## [Decision Letter · Decision Letter 0]

22 Sep 2023

Dear Dr GOULARD,

Thank you very much for submitting your manuscript "Emergent spatial goals in an integrative model of the insect central complex" for consideration at PLOS Computational Biology. As with all papers reviewed by the journal, your manuscript was reviewed by members of the editorial board and by several independent reviewers. The reviewers appreciated the attention to an important topic. Based on the reviews, we are likely to accept this manuscript for publication, providing that you modify the manuscript according to the review recommendations.

Sincerely,

Joseph Ayers, PhD

Academic Editor

PLOS Computational Biology

Daniele Marinazzo

Section Editor

PLOS Computational Biology

Reviewer's Responses to Questions

**Comments to the Authors:**

Reviewer #1: This is a very interesting computational/theoretical study that proposes a novel function for path integration in the insect central complex (Cx). While this well-documented behavioral capacity has previously been shown to allow central place foraging insects to return to their nests, the authors here propose that computation of a path integration vector could be combined with goal direction encoding to allow the insect to estimate the location of a goal in an allocentric framework. The basic idea (if I understand correctly) is that by combining internal representations of two vectors— one pointing towards a “home location” and one an estimate of the goal direction— an insect can estimate the location of a goal without the kind of place map observed in the vertebrate hippocampus. The authors demonstrate the plausibility of this scheme by building on their previously developed model of path integration in the Cx, and show that it improves navigational performance in two visually-guided tasks. Overall this is a fascinating and important new idea in insect navigation that makes several testable hypotheses. My comments largely have to do with clarifying presentation of some of the more difficult concepts (and checking if I have understood them correctly).

I had a couple of suggestions to enhance the clarity of the exposition:

1) Figure 1 dives already into complex anatomy of the insect brain. Perhaps it would be useful to introduce the main mathematical concept (in terms of vector representation and addition) as in Fig. 7 F near the top of the paper for readers who are less familiar with CX anatomy?

2) Model exposition: this section was a bit tough to read and I wonder if it can be made a little easier on the reader. For example, each of sections 4.2.2, 4.2.3, and 4.2.4 begin by referencing previous work and then go straight to names of circuit elements followed by equations. I wonder if it would clearer for readers who are not familiar with this earlier work to start each section with a conceptual overview of what that part of the circuit does (e.g. for section 4.2.2., one could start with the material at line 175: “This circuit transforms the signal from a single activity bump…”) then describe the components of the circuit and how they interact in words, then show the equations. The most challenging part of the manuscript for me was 4.2.4, but I thought a nice conceptual overview was given in the legend to Fig. 3C. Could this be brought to the top of the section and then unpacked to show how it is implemented in the model? Also in this section, could the authors spell out a bit more clearly what role the ßPFN and ßh∆ parameters play? It would be helpful here again to have a conceptual overview before the parameters are introduced.

a few typos or grammar fixes:

line 240: the term that modulate

line 446: Despite, while both model show

A few questions for the Discussion:

— Do I understand correctly that in this study the vector memories are stored in FBt synapses but have their effect by acting on connections from h∆ or PFN to PFL? It seems that some FBt neuron target axonal regions of h∆ cells and some target dendritic regions. What might be the function of those targeting dendritic regions?

— line 717-719 notes that in this circuit a sensory memory cannot be recalled based on the PI state. However, the connectome does contain “FS” neurons that project from the FB back up to regions near the output of the MB. These have not been much studied but could be a basis for additional interactions between the spatial positioning system and the sensory memory system.

Reviewer #2: This paper seems to me a very worthwhile exploration of how exhaustive modelling of the currently known circuitry in the central complex can account for a wide array of different navigational properties. It emphasises in particular the enormous importance of path integration in insect navigation and is well suited for publication in PlosS Computational Biology. I am a biologist rather than a computational modeler. On the assumption that many readers will also be biologists, I can best help by pointing out where the paper is a little difficult to follow. The earlier PLOS Computational Biology in 2021 paper by some of the same authors (ref 26) was more reader friendly.

ABSTRACT

Line 8: ‘This transforms’ is a bit unclear, ‘Path integration transforms’ might be better.

Line 10: ‘across insect species’ could be broader and include crustaceans, perhaps arthropod – a term that is used later.

AUTHOR SUMMARY

Line 12: 2 or 3 dimensional spatial problems 3 dimensional doesn’t come up elsewhere so seems odd to have it in the summary (only examples that I know come from spiders, jumping spiders do 3D PI D.E.Hill and Portia M.Tarsitano)

Line 13: In this paper, we modelled a neural pathway that sustains insect visual-guided navigation both to a ‘might neural circuitry be clearer?

INTRODUCTION

Great first para.

Lines 10-11:Perhaps reword: ‘high structural conservation across species [10, 11]’ to and its structure is highly conserved across species ?

Lines 42 -48: Passage below is quite hard to follow in an introduction. Refer to Fig 1 in the Intro?

‘In addition to this positioning system, the projection geometry

of some CX neurons effectively permits mental rotation of directional inputs. The virtual 180° shift

carried by hΔb cells subserves the transformation of the allocentric orientation, from the head direction

circuit, into an egocentric representation of the insect’s holonomic motion [16, 17].’

Line 52 ‘efficient trap-lines between multiple feeders’ Explain trap-lines (rewarded locations visited in a set order)?

Line 57 Say a little more about central place foragers – social insects like ants and bees that live in a nest and from which foragers etc.

Lines 66-73 do not read well. This section is also the first time FB is mentioned with no description or fig

MODEL

Lines 88-90 . It might help to give a brief account of ‘ring attractor’ and ‘bump of activity’.

Figure 1 is excellent but has no abbreviations so in the legend give full wording with abbreviations in parenthesis, e.g. Ellipsoid body (EB)

Line 94 Explain how inhib neurones (ER), as shown in Fig 1, carry directional info

100-103 It makes sense but would be easier to follow if it were unpacked a bit.

Fig 5A Probably my lack of insight but I don’t get panel C.

Fig S2 legend should E be D?

Line 382. Is ref 71 good evidence for degradation of route learning in CX. The claim in the abstract is ‘Thus, CX lesions had a specific impact on learnt visual guidance’ In the experiments the route is implemented by facing in the right visual direction and then being guided by a PI memory. So isn’t break-down expected because of interference with PI. Shouldn’t one therefor examine whether the lesions disrupt where the ant faces at the very start of its path?

Incidentally, in the same vein, I like the modelling of how PI can aid route following.

**Have the authors made all data and (if applicable) computational code underlying the findings in their manuscript fully available?**

Reviewer #1: Yes

Reviewer #2: **No: **Dataset will be provided on demand

PLOS authors have the option to publish the peer review history of their article (what does this mean?). If published, this will include your full peer review and any attached files.

Reviewer #1: No

Reviewer #2: No

Figure Files:

Data Requirements:

Reproducibility:

References:

---

## [Editor Report · Decision Letter 1]

1 Dec 2023

Dear Dr GOULARD,

We are pleased to inform you that your manuscript 'Emergent spatial goals in an integrative model of the insect central complex' has been provisionally accepted for publication in PLOS Computational Biology.

Best regards,

Joseph Ayers, PhD

Academic Editor

PLOS Computational Biology

Daniele Marinazzo

Section Editor

PLOS Computational Biology

---

## [Editor Report · Acceptance letter]

12 Dec 2023

PCOMPBIOL-D-23-01402R1 

Emergent spatial goals in an integrative model of the insect central complex

Dear Dr GOULARD,

I am pleased to inform you that your manuscript has been formally accepted for publication in PLOS Computational Biology. Your manuscript is now with our production department and you will be notified of the publication date in due course.

With kind regards,

Judit Kozma
